



# Reviews and Syntheses: Ironing Out Wrinkles in the Soil Phosphorus Cycling Paradigm

Curt A. McConnell[1], Jason P. Kaye[2], Armen R. Kemanian[1]

[1]Department of Plant Science, The Pennsylvania State University, University Park, PA, 16802, USA
[2]Department of Ecosystem Science and Management, The Pennsylvania State University, University Park, PA, 16802, USA

*Correspondence to*: Armen R. Kemanian (kxa15@psu.edu)

**Abstract.** Soil phosphorus (P) management remains a critical challenge for agriculture worldwide, and yet we are still unable to predict soil P dynamics as confidently as that of carbon (C) or nitrogen (N). This is due to both the complexity of inorganic P (Pi) and organic P (Po) cycling and the methodological constraints that have limited our ability to trace P dynamics in the soil-plant system. In this review we describe the challenges to building parsimonious, accurate, and useful P models and to explore the potential of some new techniques to advance modeling efforts. To advance our understanding and modeling of P biogeochemistry, research efforts should focus on the following: 1) update the McGill and Cole (1981) model of Po mineralization by clarifying the role and prevalence of "biochemical" and "biological" Po mineralization which we hypothesize are not mutually exclusive and may co-occur along a continuum of Po substrate stoichiometry; 2) further understand the dynamics of phytate, a 6-C compound that can regulate the poorly understood stoichiometry of soil P; 3) explore the effects of C and Po saturation on P sorption and Po mineralization; and 4) resolve discrepancies between hypotheses about P cycling and the methods used to test these hypotheses.

## Introduction

Managing agricultural soil phosphorus (P) to maximize crop productivity and minimize P pollution is one of the critical challenges of contemporary agriculture. Our current understanding of soil P cycling lags carbon (C) and nitrogen (N) (Reed et al., 2015), and that lag is pronounced in simulation models (e.g. Vadas et al., 2006). Most P cycling models are structured after C and N models despite key differences between the biogeochemical cycling of the three elements. Soil P has a significant inorganic component in addition to an organic one, both of which can cycle independently of C and N (Condron et al., 2005; McGill and Cole, 1981). Unlike C and N, investigating P cycling is limited by the existence of one P stable isotope (C and N have two each), making tracing studies dependent on P radioisotopes that are short-lived and hazardous, or on phosphate-bound stable oxygen isotopes that are subject to biotically-mediated fractionation, complicating the interpretation of abiotic P transformations (Angert et al., 2011; Blake et al., 2005).

Gaps in our understanding of the P cycle are reflected in model conceptualizations and in model results that are rather uncertain regarding P (Radcliffe et al., 2009). The mismatch or poor correlation between analytical-chemical P pools and conceptual P pools (Gijsman et al., 1996), only compounds this uncertainty. A manifestation of these limitations in model prediction was the failure to predict increased soluble P losses as agricultural management shifts from till to no-till systems (Duncan et al., 2019), even though it was long known that while no-till reduces erosion it stratifies immobile nutrients. Vertical stratification of P coupled with hydrological connectivity between fields and waterways via runoff or tile drains has led to enhanced transport of P from fields to water (Dodd and Sharpley, 2016; Radcliffe et al., 2015). The collective inability to predict the behavior of P with current nutrient models (García et al., 2016; Radcliffe et al., 2015) highlights the limited holistic understanding of P cycling.



The goal of this review is to identify gaps in soil P cycling research that contribute to the observation-model disconnect and to clarify the path forward for a more seamless integration of experimental and theoretical progress. The objectives are to review the (1) discrepancies in Po mineralization paradigms, (2) difficulties in incorporating phytate and C:P stoichiometry into these paradigms, (3) dynamics of C and Po saturation, and (4) methodological challenges in measuring P content, distribution, and sorption. These specific subjects were selected based on the extent of diverging evidence and the degree of importance to the conceptualization, or reconceptualization, of soil P cycling.

**Overview of P cycle**

In the soil-plant system, P exists in both inorganic (Pi) and organic (Po) forms whose relative abundance depends on edaphic, management, and climate conditions. Generally, Po makes up 35-65%, and sometimes up to 90%, of total P in soil (Harrison, 1987; Williams and Steinbergs, 1958). Of the total P in most plant-soil systems, less than 10% is in living organisms (Arai and Sparks, 2007; Ozanne, 1980). Usually less than 1% is in solution as either a dissolved organic species or as a free inorganic phosphate ($PO_4$) ion immediately useable by plants and microorganisms (Frossard et al., 2000; Pierzynski, 1991). For $PO_4$ to become available in the soil solution for organism uptake, soil Pi and soil Po must undergo abiotic and biotic transformations (Figure 1).

Soil Pi is either in a primary mineral form, precipitated as various inorganic phosphates (mainly as $AlPO_4$, $FePO_4$, $CaPO_4$), occluded by precipitates and other minerals, sorbed to mineral surfaces, or dissolved in the soil solution. In inorganic "physiochemical" pathways, $PO_4$ is released into solution by the dissolution of minerals, precipitates, and sorbed Pi (Arai and Sparks, 2007; Cross and Schlesinger, 1995; Hedley et al., 1982). Due to $PO_4$'s strong affinity to mineral and colloid surfaces, $PO_4$ equilibrium favors sorption to the solid phase rather than desorption to the soil solution (Menezes-Blackburn et al., 2016; Okajima et al., 1983).

Once sorbed to minerals by electrostatic interaction or fast chemisorption (ligand exchange on external mineral sites) $PO_4$ is either desorbed back to the soil solution or stabilized through further ligand exchange and slow incorporation into aggregates or clays (Arai and Sparks, 2007; Holtan et al., 1988). The methods of Hedley et al. (1982), later refined by Tiessen and Moir (1993), attempted to quantify P into pools along this continuum of P stabilization using sequential chemical extraction methods. The so-called "Hedley fractionation" is the most widely used method to quantify both Pi and Po in fractions of varying lability and indirectly, plant availability (Guppy et al., 2000).

The dissolution of solid phase Pi, and thus plant availability, depends on the ecosystem type (Bünemann, 2015; Cleveland et al., 2013), degree of soil development (Smeck, 1985), parent material (Bünemann et al., 2016), Pi lability relative to Po (Pistocchi et al., 2018), and human inputs (Oehl et al., 2004). Soils in early stages of development have low soil Po levels so plant P acquisition relies on physiochemical processes for Pi dissolution. These physiochemical processes also tend to control P availability for plants in agroecosystems supplied with Pi-rich fertilizers (Bünemann, 2015; Guo et al., 2000; Oehl et al., 2004). While Po can play a less significant role in agricultural systems (Bünemann, 2015), it can be an important P source via mineralization in forested or highly weathered soils (Bünemann, 2015; Cleveland et al., 2013; Yang and Post, 2011), particularly when labile Pi availability is low (Pistocchi et al., 2018).





The most common soil Po forms are orthophosphate monoesters, orthophosphate diesters, organic polyphosphates, and phosphonates (Turner et al. 2007). The orthophosphate monoester inositol phosphate comprises a significant fraction of total Po in most measured soils, principally due its recalcitrance to mineralization (Turner et al. 2002). Sugar phosphate monoesters and phosphate diesters like DNA and RNA are more labile sources of mineralizable Po (Condron et al., 2005; Turner, 2008).

The agents responsible for Po mineralization are plant and microbial phosphatases that are periplasmic, membrane-bound, or exuded into the soil solution. One function of extracellular enzymes is to depolymerize Po compounds, thus increasing their solubility and accessibility by microbes and plants (Quiquampoix and Mousain, 2005). After depolymerization, Po monomers may be assimilated by a microorganism depending on the presence of compound-specific membrane transport proteins and other organismal and environmental controls such as the concentration of Pi or the Po compound (Heath, 2005; Quiquampoix and
Mousain, 2005; Wanner, 1996). However, only a few Po compounds are known to have direct transmembrane mobility (Wanner, 1996) and to our knowledge studies of direct Po uptake were not conducted in soils. The indirect uptake of $PO_4$ from non-transportable Po compounds is regulated by phosphatase enzymes.

The mechanisms by which non-transportable Po is mineralized by microbes and plants are not clearly described in the literature; there is no unifying principle of Po mineralization considering recent advances in P biogeochemistry research. Initially,
microbial Po mineralization was described similarly to N mineralization, whereby Po is mineralized in conjunction with C for energy, and therefore coupled to C dynamics (Dalal, 1979; Thompson et al., 1954). This "biological" Po mineralization model was challenged by McGill and Cole 1981, who posited a "biochemical" mineralization pathway. Under their definition, Po mineralization is driven by P demand and enabled by phosphatase enzymes to acquire P independent of C and is therefore decoupled from C dynamics. Although this conceptualization has been generally accepted to occur under low labile Pi conditions
and provides a potential explanation for the observed variability in soil and microbial nutrient stoichiometry, it is not complete (see next section).

The biological and biochemical pathways are generally regarded as mutually exclusive; but it is simpler to assume that they coexist. Po mineralization can result in acquisition of P, C, or both, depending on the relative demand for either element or the relative lability of the Po substrate. The biological and biochemical terms are misleading as Po mineralization is always
biochemical, i.e. phosphatase enzymes are used regardless of whether mineralization is driven by C or P demand. We propose to substitute "C-driven Po mineralization" for biological and "P-driven Po mineralization" for biochemical to distinguish the drivers, but not the paths, leading to Po mineralization. We understand that C-driven Po mineralization as a concept might be narrow in scope, as organic matter can be decomposed to mine organic N or sulfur as well, but the proposed language is unequivocal with respect to P.

**1 Deconstructing Po mineralization and updating the McGill and Cole conceptual model**

McGill and Cole (1981) proposed that Po mineralization is independent of C and N mineralization (Smeck, 1985). However, this model is not uniformly true, as elucidated by Condron et al. 2005 and further evidenced by recent research. Phosphatase enzymes play a key role in Po mineralization, yet phosphatase-mediated dephosphorylation may also be a requisite step in the acquisition of C (Spohn et al., 2015; Wang et al., 2016) as the polar $PO_4$ groups may hinder direct Po uptake (Steenbergh et al., 2011). The
extent to and conditions under which either C-driven or P-driven Po mineralization occur is debated and requires clarification (Gressel et al., 1996; Guppy and McLaughlin, 2009; Heuck and Spohn, 2016; Smeck, 1985; Spohn and Kuzyakov, 2013).





### 1.1 Evidence supporting P-driven Po mineralization

Phosphatase enzymes can mineralize Po independent of the C moiety. There is substantial evidence relating Po depletion to increased phosphatase activity (Asmar et al., 1995; Chen et al., 2002; Häussling and Marschner, 1989; Rojo et al., 1990; Speir

and Cowling, 1991; Tarafdar and Jungk, 1987) suggesting a mineralization pathway driven by P demand and decoupled from C dynamics. Increased phosphatase activity is also observed when decomposition is limited by P availability (Sinsabaugh et al., 1993), the soil has a high organic carbon (OC) content (Halstead, 1964), or organic material is added to a soil (Li et al., 2015). Furthermore, there is a general negative relationship between available Pi and phosphatase activity, suggesting that P-driven mineralization is suppressed when labile Pi is no longer limiting (Clarholm, 1993; Colvan et al., 2001; Dick et al., 1988; Juma

and Tabatabai, 1977, 1978; Marklein and Houlton, 2012; Moscatelli et al., 2005; Nannipieri et al., 1978; Spiers and McGill, 1979; Turner and Joseph Wright, 2014).

A tighter coupling of C mineralization with soil organic matter (SOM) C:N ratios rather than C:P ratios (Heuck and Spohn, 2016), and a relatively poor correlation of Po with C or N (Yang and Post, 2011), indicate a greater independence of P mineralization from C than does N. Furthermore, phosphatase activity responds quicker to changes in mineral P availability than

analogous N enzymes to N availability (Marklein et al., 2016) indicating the reliance on phosphatase enzymes for P acquisition under P-limited scenarios.

### 1.2 Conflicting evidence in discerning P-driven and C-driven mineralization

The relative contribution of plants and microbes to exuded phosphatase enzymes is unknown (Richardson et al., 2009b). Because plants acquire C through photosynthesis, P-driven Po mineralization seems uncontroversial for autotrophic plants. In contrast,

heterotrophic microbes may mineralize Po compounds for C or P rather than solely P. Discriminating microbial and plant phosphatase activity in the rhizosphere and linking C- or P-driven Po mineralization pathways to either organism is challenging.

Phosphatase activity is higher in the rhizosphere than in the bulk soil (Häussling and Marschner, 1989; Nannipieri et al., 1978) likely due to the greater abundance of Po substrates from decomposing root and microbial matter and combined plant and microbial activity near roots (Jones et al., 2009; Spohn et al., 2015). The rhizosphere also tends to exhibit lower phosphatase

activity with higher Pi availability compared to the bulk soil (Hedley et al., 1983). This response cannot be conclusively linked to plants or microbes (Richardson et al., 2009b) because, with a few exceptions, plant and microbial phosphatases are indistinguishable (Nannipieri et al., 2011).

Among the exceptions are the microbe-specific alkaline phosphatases, which in some studies have exhibited a negative response to Pi fertilization when measured in the rhizosphere (Spohn et al., 2015). This indicates active microbe-mediated P-driven Po

mineralization. It must be noted however that alkaline phosphatases are only a subset of the microbially-produced phosphatase enzymes and are produced in neutral to basic conditions (Juma and Tabatabai, 1978), which cautions against broad generalizations.

Outside of the rhizosphere in the bulk soil, it has been shown that the alkaline phosphatase activity response to Pi availability is subdued (Spohn et al., 2015) or even positive (Colvan et al., 2001). This pattern is consistent with C-driven Po mineralization by

non-rhizosphere microbes, stripping the P group as a prerequisite for further C processing (Spohn et al., 2015; Spohn and Kuzyakov, 2013). Supporting this interpretation, bulk or incubated soil isolated from roots (with phosphatases more likely of





microbial origin) have not shown a decline in either acid or alkaline phosphatase activity when Pi availability increases (Adams and Pate, 1992; Lima et al., 1996).

Additionally some studies have either found no relationship between Po mineralization and phosphatase activity (Allison and Vitousek, 2005; Chen et al., 2000; Dissing Nielsen and Eiland, 1980; Trasar-Cepeda et al., 1991) or did not observe a negative relationship of phosphatase activity with Pi availability or Pi fertilization (Halstead, 1964; Harrison, 1983; Li et al., 2015). The lack of an immediate negative response to Pi availability may be caused by recalcitrant phosphatase enzymes stabilized by soil minerals and colloids that remain active in the soil (Clarholm, 1993; Turner and Joseph Wright, 2014). Turner and Wright (2014) proposed that longer term studies must be conducted to ensure that a change in overall phosphatase production can be observed. But an equally plausible explanation for a lack of a phosphatase response to Pi availability may be that microbes are releasing phosphatase as a means of C acquisition (Heuck et al., 2015; Spohn and Kuzyakov, 2013), so phosphatase production can be on occasion unrelated to increased Pi availability (Figure 2).

### 1.3 Evidence supporting C-driven Po mineralization

Evidence for C-driven Po mineralization comes from studies showing the coupling of Po mineralization and SOM decomposition (Dalal, 1979; Gressel et al., 1996; Moller et al., 2000). In dual-labeling experiments with $^{33}P$ and $^{14}C$, the preferential microbial uptake of C from labeled glucose phosphate was greater than that of P, even under C saturation (Heuck et al., 2015; Spohn and Kuzyakov, 2013). Fransson and Jones (2007) found that organic compounds like adenosine were preferentially mineralized over their phosphorylated counterparts (AMP, ADP, ATP), and that low phosphatase activity may limit the use of phosphorylated compounds at elevated Po concentrations. Accordingly, the activity of the C mineralizing enzyme β-glucosidase explained 41 to 69% of the variation in phosphatase activity (Wang et al., 2016) indicating the two groups of mineralization enzymes can be closely related (Sinsabaugh et al., 2009).

Although poorly understood and undertested in soils, dephosphorylation of Po compounds may be such an important step in C acquisition from Po because soil microbes may be limited in their ability to directly assimilate Po compounds in the soil. So far there is limited to no evidence of direct Po assimilation in soils (George et al. 2017).

### 1.4 The co-existence of C-driven and P-driven Po mineralization

It may only be on the extremes of a continuum of C or P limitation where C- or P-driven Po mineralization dominates (Figure 3) with co-existence of these mechanisms under co-limiting conditions. In a temperate P-limited system, Heuck et al. 2015 found that C-driven Po mineralization dominated, likely due to concurrent C-limitation that drove mineralization. Similarly, different microsites in the soil can be P- or C-limited, further complicating data interpretation when analyses are done in soil volumes that cannot distinguish processes at microscales.

Strict P-driven mineralization may not apply to soils that are C-limited, but many soils are not C-limited (Zechmeister-Boltenstern et al., 2015), particularly in tropical systems that can have significant P limitations (Camenzind et al., 2018). In highly-weathered soils, the strong Pi fixation potential and overall low Pi makes Po the dominant P source in the soil (McDowell et al., 2007; Vincent et al., 2010), and thus P-scavenging P acquisition strategies may be more prevalent. Soil type does not completely explain the dominance of P-driven Po mineralization, as it may also depend on Po speciation. Vincent et al. 2010 observed that litter manipulations did not impact soil C:N:P stoichiometry in a lowland tropical forest, indicating a non-



discriminating mineralization behavior. As noted by these authors, the observed absence of inositol phosphates, which are very recalcitrant Po compounds, may have reduced the prevalence of P-driven mineralization. A depletion in inositol phosphate, as seen in other studies (McDowell et al., 2007), may have been caused by earlier P-driven Po mineralization (McDowell et al., 2007; Turner et al., 2007). Organic substrates of varying composition and their varying propensity to specific mineralization pathways may influence P stoichiometry and Po mineralization.

**1.5 Synthesis**

The co-regulation of Po mineralization by substrate properties and by plant and microbial P-demand needs to be clarified. Developing methods to measure real instead of potential phosphatase activity will allow a more accurate estimation of Po mineralization from actively produced phosphatase enzymes (Bünemann, 2015; Nannipieri et al., 2011). This will reduce the overestimation problem of current *in-vitro* phosphatase enzyme assays (Nannipieri et al., 2011; Spohn et al., 2013), and may enable more accurate parameterization of models based on phosphatase activity such as that of Schimel and Weintraub (2003). However, this is clearly not enough.

Microbial or plant demand for P can also regulate phosphatase production, and the properties of Po would determine the effectiveness of these phosphatases. A typical approach in modeling is to determine the maximum turnover rate of a given pool, which in this case can be equated to the rate of Po mineralization in systems that are flooded with phosphatase. This rate can be reduced by the actual phosphatase activity, which in turn would depend on a calculated demand from microbes and plants. Co-located organic C mineralization would also influence the Po turnover rate. To our knowledge, no systematic effort has been conducted to elucidate such relationships.

P turnover and the linkage with organic C mineralization can be studied with $^{32}$P and $^{33}$P isotope dilution (Frossard et al., 2011) and $^{18}$O-P tracing to obtain mineralization rates alongside enzyme activity assays. $^{18}$O isotope P tracing techniques has been used as a safe alternative to radioisotopes in tracing inorganic phosphate (Tamburini et al., 2010, 2014), and recent methods have demonstrated the efficacy of $^{18}$O tracing of organic phosphate (Liang and Blake, 2006; Tamburini et al., 2018). However, Po tracing methods using oxygen isotopes are in their infancy and have not been applied outside of a proof-of-concept. Experiments employing $^{18}$O tracing techniques for both Pi and Po must be conducted on a wider range of soils (Nannipieri et al., 2011). Further employment of these methods could help discern biological processes and the turnover of phosphate in natural abundance (Roberts et al., 2015) and tracer studies (Joshi et al., 2016). They can also be used for P source characterization (Frossard et al., 2011; Zohar et al., 2010).

Discriminating the activity of microorganisms from those of stabilized enzymes is also tractable (Turner and Joseph Wright, 2014). One way would be to measure the expression of phosphatase genes in plants and microorganisms as a response to changes in P availability. A characterization of microbial or plant responsiveness to shifts in P availability would further help integrating P-driven Po mineralization into models, as it is determined by specific environmental and biological conditions. This requires further work identifying conditions that lead to changes in gene expression (Grafe et al., 2018).

Most models do not explicitly simulate P-driven Po mineralization independent of C demands, (Reed et al., 2015) largely because the general structure of P models mirrors that of N, which only includes C-driven Po mineralization. The global CNP models CLM-CNP and CASA-CNP incorporate P-driven Po mineralization of SOM pools but there are insufficient observations of this process for reliable parameterization (Achat et al., 2016; Reed et al., 2015; Wang et al., 2010; Yang et al., 2014).





Although explicit inclusion of P-driven Po mineralization resulted in better agreement between simulated and measured P pool
sizes (Wang et al., 2010), P-driven Po mineralization may only apply in specific circumstances (Olander and Vitousek, 2000).

With major uncertainties in theory, there are limitations to venture far from data.

## 2 Phytate dynamics and the unpredictability of P stoichiometry

### 2.1 Phytate

Po species differ in abundance due to their variable affinity for mineral or SOM sorption and recalcitrance to mineralization.
Commonly, the most prevalent Po forms in soil are inositol phosphates consisting of six-carbon rings with one to six phosphate

groups. Myo-inositol hexakisphosphate, also known as phytate in its salt form, is the most abundant inositol phosphate and has
six P groups (Harrison, 1987; Turner et al., 2002). Phytate is stabilized in the soil through ligand exchange, formation of metal-
phosphate bridges to SOM, and precipitation as insoluble salts (Celi et al., 1999; Jørgensen et al., 2015). The phosphate groups
bind to mineral surfaces by ligand exchange with hydroxyls and to mineral and SOM cationic surfaces via electrostatic binding
(Arai and Sparks, 2007; Celi et al., 1999; Jørgensen et al., 2015). Phytate's tendency to bind to the soil phase contributes to its

abundance and recalcitrance to mineralization (Anderson et al., 1974; Berg and Joern, 2006; Turner et al., 2002). A majority,
from 29 to 65% and even up to 90%, of soil Po can be phytate (Harrison, 1987; Turner et al., 2002) but this can vary
substantially depending on factors that are not fully understood (Figure 4).

A thorough review on inositol phosphates is found in Turner (2007), but relevant information and new findings will be covered
here. Phytate is prevalently stored in grain (and pollen, which is a small pool) but is also found in roots, crowns, and leaves,

potentially as a transient storage compound (Campbell et al., 1991; Hubel and Beck, 1996). Turner (2007) noted that despite
prevalent research on phytate and other soil inositol phosphates, their abundance varies widely and often unpredictably. As
described by McGill and Cole (1981) and later supported by Shang et al. (1990), phytate sorption dynamics is similar to that of
orthophosphate and is partially controlled by similar factors (sorption capacity, Fe and Al oxide content) (Yan et al., 2014). The
abundance of phytate measured in agricultural systems is attributed to their recalcitrance and stability through sorption and

precipitation (Yan et al., 2014) and prevalence in manures (particularly non-ruminants) and animal feeds (Sun and Jaisi, 2018).

#### 2.1.1 Controls on phytate mineralization

The solubility and lability of phytate compounds, the principal controls on phytate mineralization, and the extent and efficacy of
its mineralization by plants remain unclear (Gerke, 2015a; Richardson et al., 2000). We do know that phytate mineralization is a
two-step process whereby phytate is first solubilized and made accessible to phytate-hydrolyzing enzymes called "phytases"

(Gerke, 2015b, 2015a; Mullaney and Ullah, 2007). In this review, the term phytase is used rather than the more general term
"phytate-degrading enzymes", which refers to any enzyme *in vivo* or *in vitro* that can hydrolyze phosphate from phytate
(Greiner, 2007). Because this review focuses on the *in vivo* processes, the term phytase is preferred.

The two principal controls on phytate mineralization are the production of phytase and the solubility of phytate, but their relative
influence on mineralization is not fully understood. Phytate is expected to undergo hydrolysis in the soil solution (Ognalaga et

al., 1994), but for that to happen it must first be solubilized from the soil phase through abiotic or biotic processes (Gerke, 2015b,
2015a). Akin to other organic phosphates, biotically-mediated dissolution of phytate can be facilitated by organic anions that
chelate Fe, Al, or Ca (Tang et al., 2006). While solubilization is thought to be the limiting step in the hydrolysis of phytate





(Gerke, 2015a; Greaves and Webley, 1969; Lung and Lim, 2006; Patel et al., 2010), other studies have shown that hydrolysis and phytase production are also limiting steps (Findenegg and Nelemans, 1993; George et al., 2004; Hayes et al., 2000; Richardson et
al., 2001).

Microorganisms are key regulators of phytate mineralization. They produce both phytate-solubilizing organic acids (Richardson and Simpson, 2011) and multiple classes of phytate mineralizing enzymes (Hill and Richardson, 2007; Mullaney and Ullah, 2007). Soil microbial communities appear to be crucial agents in increasing phytate availability for plant acquisition (Richardson and Simpson, 2011). The extent to which plants themselves control phytate mineralization outside of in-vitro studies or genetic
modification is less clear.

### 2.1.2 Plant-mediated phytate mineralization

A limited number of reports indicate that some plant species can secrete phytase (Belinque et al., 2015; Li et al., 1997). This is likely a response to limited Pi availability (Tarafdar and Claassen, 2003), and would be a prime example of P-driven Po mineralization with phytate as the substrate. Steffens et al. (2010) demonstrated plant-mediated phytate mineralization in a
growth chamber experiment with negligible microbial activity. While studies of this nature are scarce, Belinque et al. (2015) found that oilseed rape, sunflower, and soybean grown under sterile conditions could use phytate as a P source and that microbial inoculation had a minimal effect on overall plant acquisition of phytate P.

Contrary to these findings, other authors have reported that phytate mineralization and subsequent uptake of phytate-derived P by plants is minimal or absent (Findenegg and Nelemans, 1993; Lung and Lim, 2006; Richardson et al., 2000, 2001) or that detected
phytase was plant-derived but involved in intracellular root phytate regulation rather than extracellular phytate scavenging (Asmar, 1997; Hubel and Beck, 1996; Richardson et al., 2000). If extracellular phytase release from plants is truly minimal or absent, effective plant use of soil phytate would depend heavily on phytase-producing microorganisms (Idriss et al., 2002; Richardson et al., 2000) or expression of microbial phytase genes in transgenic plants (Giles et al., 2017; Lung et al., 2005; Richardson et al., 2001). Furthermore, phytate use may still be limited by either microbial exoenzyme production (Findenegg
and Nelemans, 1993) or phytate accessibility (Gerke, 2015a, 2015b).

### 2.1.3 Variable abundance and mineralization rates of soil phytate

There are exceptions that do not fit, or question, a neat pattern of phytate stabilization and retention in soil. Contrary to the widespread abundance of phytate often attributed to its recalcitrance to mineralization, rapid phytate mineralization has been observed in non-calcareous (Dou et al., 2009) and calcareous soils (Doolette et al., 2010; Leytem et al., 2006). Possible
explanations to rapid mineralization are that certain soil conditions increase phytate solubility or that some phytates are inherently more soluble or susceptible to hydrolysis depending on interactions with specific minerals or SOM.

As explained by Turner and Blackwell (2013), unless the soil solution of a calcareous soil has excess dissolved $Ca^{2+}$, Ca-phytate is slightly more soluble than Fe/Al phytates (Jackman and Black, 1951). Furthermore, Ca-phytate mineralizes more rapidly than insoluble Fe or Al-associated phytates (Greenwood and Lewis, 1977; Quiquampoix and Mousain, 2005; Tang et al., 2006) even
at a pH 6-8 (Greaves and Webley, 1969)

Working in a non-calcareous soil, Dou et al. 2009 found a lack of phytate accumulation and presumed rapid mineralization. In this case, although Al and Fe contents and fixation capacity may have been high, the soil was likely saturated with OC from





previous manure applications. This would have increased the solubilization of phytate and contributed to its observed rapid mineralization, a subject discussed further below in the "Organic P and C saturation" section.

**285  2.1.4 Synthesis**

Although both plants and microbes can release phytase and phytate-solubilizing organic acids (Richardson et al., 2009b), microbes indirectly facilitate phytate acquisition (Richardson et al., 2001, 2009a) likely due to greater production of phytate solubilization or mineralization enzymes. It is therefore necessary to further investigate the interplay between microbes and plants, particularly *in vivo*, where experimentation is limited (Giles and Cade-Menun, 2014).

**290**  Although phytate makes up a large percentage of soil Po in many soils, its dynamics are not explicitly simulated in models. Instead of constructing general pools with presumed Po turnover rates, specific forms of C and P should be identified and their roles in the turnover of Po investigated (Arenberg and Arai, 2019) with the ultimate goal of incorporating phytate-specific pools in models, such as that depicted in Figure 5. However, this will require elucidating the actual phytate abundance, as it is likely overestimated due to limitations in phytate analysis (Doolette et al., 2011; Smernik and Dougherty, 2007).

**295**  Integrating phytate pools into models would be facilitated by further research on the sorption of phytate and its mobility (Gerke, 2015b), the complementary effects of organic acid and phytase exudation from both plants and microbes on phytate depletion (Giles et al., 2017), and the observations of rapid phytate mineralization (Doolette et al., 2010) and reduced stabilization (Dou et al., 2009) resulting in variable phytate abundance. Coupling isotope tracing techniques and $^{31}$P NMR spectroscopy techniques will also provide important insights into the fate of Po compounds like phytate (Giles and Cade-Menun, 2014; Tamburini et al.,

**300**  2018). However, further study on plant and microbial strategies for improving phytate acquisition must also be conducted at the field scale (Giles and Cade-Menun, 2014). There is also a need to eliminate or reduce the ambiguities in $^{31}$P NMR techniques when characterizing or quantifying Po, a topic covered in Kruse et al. (2015). The lack of clarity surrounding the different mineralization pathways, the varying recalcitrance of Po species, or a combination of the two may contribute to the observed wide global variation in C:P or N:P stoichiometry (Tipping et al., 2016; Vincent et al., 2010).

**305  2.2 Stoichiometry**

The flux of nutrients between soil and organisms can be modeled by tying C fluxes to C:N:P stoichiometry, if it is known and predictable. However, variation in soil C:P ratios, flexible microbial stoichiometry, and unpredictable microbial critical ratios (CRs) contribute to model uncertainty. Variability in soil and microbial stoichiometry derive from methodological or analytical discrepancies (Kirkby et al., 2011), edaphic and ecosystem properties, and characteristics specific to microorganisms (Čapek et

**310**  al., 2016).

**2.2.1 Soil Stoichiometry**

Knowledge of C:N ratios has enabled accurate modeling of N limitation and mineralization-immobilization dynamics (White et al., 2014). This modeling success can be attributed to the tight coupling of C and N in soils across ecosystems, which is largely because organic N makes up 95% of soil N (Duxbury et al., 1989; Kirkby et al., 2011; Yang and Post, 2011). Soil C:P and N:P

**315**  ratios are currently used in simulation models, but unlike N, Pi and Po are often poorly correlated with soil C or N content (Hartman and Richardson, 2013; Tipping et al., 2016; Yang and Post, 2011; Zhou et al., 2018).





The use of Po, Pi, or Pt in stoichiometry measurements requires clarification as methodological differences (Kirkby et al., 2011) can confuse matters. Including Pi in C:Pt (total P) ratios introduces more variability in relating C and P because Pi can cycle

independent of C and N. The C:Po ratio, where C only includes organic C, may better represent Po stoichiometry in the soil, but it too can vary widely depending on the plant and microbial communities, ecosystem, and management (Figure 6) (Čapek et al., 2016, 2018; Hartman and Richardson, 2013; Tipping et al., 2016). The decoupling of C:Pt and N:Pt is also seen as soil weathers (Yang and Post, 2011) where Po becomes the predominant contributor to P fertility (Bünemann, 2015; Cleveland et al., 2013; Yang and Post, 2011). A depletion of mineral Pi, an absence of a strong C limitation, or a potential shift to P-driven Po

mineralization processes, may explain this decoupling.

### 2.2.2 Plant and Microbial Stoichiometry

The variability seen in soil C:P stoichiometry is also seen in C:P ratios of plants and microbes across and within ecosystems (Čapek et al., 2016; Cleveland and Liptzin, 2007; Hartman and Richardson, 2013; Xu et al., 2013). Plant stoichiometry is an important control on the flux of soil nutrients as it influences the decomposition activity of microbes (Manzoni et al., 2008,

2010). Microbes are the principal decomposers of litter and SOM, and therefore a major driver of P transformation in the soil (Zechmeister-Boltenstern et al., 2015), which is thought to be controlled by their own stoichiometry (Hall et al., 2011).

Microbial stoichiometry is more constrained than that of plants (Arenberg and Arai, 2019; Xu et al., 2013) and is often purported to be strictly homeostatic (Cleveland and Liptzin, 2007). Although microbial stoichiometry is roughly constrained on the global level, microbes at the population, ecosystem, or community scale may not be strictly homeostatic, as found in aquatic systems

(Cotner et al., 2010). Soil microbes may also display the same level of stoichiometric flexibility (Hartman and Richardson, 2013), varying due to population size-dependent scaling, habitat and ecosystem differences, or shifts in microbial community composition (Čapek et al., 2016; Hartman and Richardson, 2013).

Accounting for this variability in models is further complicated by the difficulties in calculating microbial nutrient demand, governed by the ratio of C:P at which microbes shift between mineralization and immobilization (Hartman and Richardson,

2013; Manzoni et al., 2010). This critical ratio (CR) is often predicted using direct measurements of microbial biomass C:P, which itself is poorly correlated to actual microbial nutrient requirements (Čapek et al., 2018). This is in part because microbes can store P in the form of polyphosphates (up to 30% of their dry weight) (Deinema et al., 1985; Kulaev et al., 1999), which reflects an indirect translation between C:P ratios and demand (Čapek et al., 2016, 2018).

The microbial $C:P_{CR}$ is an important factor to model because mineralized or immobilized substrates provide or restrict P to

plants, respectively. Capek et al. 2016 attempted to predict the $C:P_{CR}$ using microbial biomass stoichiometry and soil P measurements but were unsuccessful, likely because $C:P_{CR}$ is not solely dependent on biomass C:P, but on various edaphic and community-specific factors as well. This may explain why previous studies have failed to see a strong relationship between Po mineralization and C:P ratios of SOM (Enwezor, 1967, 1976; McLaughlin et al., 2011). Using a fixed microbial $C:P_{CR}$ may not capture the observed variability (Čapek et al., 2016; Hartman and Richardson, 2013), but implementing a flexible $C:P_{CR}$ in a

model is not yet possible due to the ratio's unpredictability (Čapek et al., 2016). However, one can tentatively assume that microbial communities adjust their P requirements by increasing $C:P_{CR}$ as P becomes more limiting. The degree of P limitation depends in part on N availability, because to sustain P-driven Po mineralization microbes need N sources to sustain phosphatase





enzyme production (Houlton et al., 2008; Olander and Vitousek, 2000). Supporting this view, Margalef et al. (2017) found that total N content was strongly correlated to phosphatase activity across global soil measurements.

**2.2.3 Synthesis**

One of the difficulties in modeling soil P dynamics is predicting the C:N:P stoichiometry of plants, microbes, and SOM. This is due to the many, often interacting, sources of variation in P stoichiometry such as habitat, edaphic properties, soil C quality, microbial population dynamics, climate, season, and disturbance or management (Aponte et al., 2010; Čapek et al., 2016; Cleveland et al., 2004; Hartman and Richardson, 2013). Establishing better relationships between these sources of variability and

observed nutrient stoichiometry as well as implementing a consistent measurement protocol for C:P ratios will help future data collection and consolidation efforts.

The variability of C:P$_{CR}$ or the microbial and C:P ratios presented in Figure 6 makes it difficult but not intractable to set constraints on P cycling models. Many models assume fixed stoichiometry for the soil or soil pools (Kemanian et al., 2011), which may only be applicable to certain soil systems. Using fixed stoichiometry in P models may not capture P dynamics across

ecosystems, but implementing variable C:P ratios, like those in the CENTURY model, requires improved parameterization and understanding of the factors that control C:P ratios. Clarifying the role of phytates may simplify this task, because phytate's C:P = 1, a ratio that is well below the C:P$_{CR}$ of any organism. It is also necessary to elucidate the mechanisms of P-driven Po mineralization because CRs do not reflect P dynamics in systems dominated by such Po mineralization pathways (McLaughlin et al., 2011).

**3 Organic P and C saturation**

P saturation refers to the level of soil P, typically inorganic, in relation to the measured sorption capacity. P saturation is often expressed as the degree of P saturation (DPS), which is used as an environmental risk indicator of potential dissolved P losses (Breeuwsma et al., 1995). Soils with DPS levels that surpass a "change point" (hereafter referred to as threshold DPS) exhibit significant increases in dissolved P transport due to the saturation of high energy sorption sites and reduction in Pi retention

strength (Abdala et al., 2012; Butler and Coale, 2005; Hooda et al., 2000; Maguire and Sims, 2002). This threshold DPS generally occurs between 25 and 56% saturation depending on the soil (Maguire and Sims, 2002).

Difficulties in narrowing the range of threshold DPS or creating a simple predictive framework can be partially attributed to the ambiguous treatment of OC saturation and Po abundance in DPS measurements. One such DPS measurement uses oxalate or Mehlich-3 extractions to obtain the molar ratio of extractable P to the sorption maximum determined by extractable Al and Fe

(Kleinman and Sharpley, 2002; van der Zee and van Riemsdijk, 1988). This method ignores the soil OC content, which if high enough, can reduce the physically obtainable P saturation potential of a soil (Table 1, Figure 7A) and influence the rate of Po mineralization (Figure 7B). Furthermore, certain extractants and analytical methods do not target Po (e.g. Mehlich-3 extraction, Pi sorption isotherms, spectrophotometric methods), leading to underestimation of the "actual" total P DPS (Table 1, Figure 7C). The extent to which DPS is affected by Po and OC depends on the relative concentrations of Pi and Po, the form of Po and its

sorption potential, and the interacting sorption relationships between Po, Pi, and dissolved organic compounds from the breakdown of SOM.





Because Po turnover and mineralization is partially controlled by its solubility (Grafe et al., 2018; Greaves and Webley, 1969), and saturation influences the solubility of both Pi and Po (Heckrath et al., 1995), ignoring Po sorption and OC saturation dynamics may limit the predictive capability of DPS measurements and sorption indices. Po and OC saturation are generally not
determined and are not explicitly simulated in models despite the potential interactions between Pi and Po in the soil and the role of Po in plant fertility.

### 3.1 Modelling the effects of OC saturation on Po mineralization

Similar to P, OC can be stabilized in the soil by sorption to silts and clays (Hassink and Whitmore, 1997) and during aggregate formation (Six et al., 2002). Sorption proceeds with the formation of organo-mineral complexes. The OC saturation ($C_x$) can be
calculated based on the clay and silt content of the soil (Table 2, Eq. 1) (Hassink and Whitmore, 1997). The saturation of OC has implications for the storage of C, decomposition, and the mineralization of nutrients (Castellano et al., 2012; Kemanian et al., 2011; Kemanian and Stöckle, 2010; White et al., 2014).

OC saturation dynamics are only included in a few models (Kemanian and Stöckle, 2010; White et al., 2014), and its effect on Po mineralization is unexplored in models. The limited number of studies on OC saturation and N cycling demonstrate that OC
saturation is positively correlated with N mineralization (Castellano et al., 2012), likely because OC saturation reduces the C transfer efficiency or rate to stable pools, thus increasing mineralization of labile N (White et al. 2014). This relationship is expected to be seen between Po mineralization and OC saturation (Figure 7B). The effects of OC saturation are most readily seen in soils with high OC accumulation (top layer of stratified, undisturbed no-till systems) (Mazzilli et al., 2014; Pravia et al., 2019), or soils with low clay content (Castellano et al., 2012). More studies are needed, especially in areas with an OC saturation
gradient, to solidify the OC saturation concept for N dynamics (White et al., 2014) and advance it for P, preferably in investigations pairing N and P.

### 3.2 Competitive OC anion sorption

Experiments on OC saturation suggest that as the OC content of the soil increases, so does the lability, solubility, and ultimate transport of Pi from the soil (Abdala et al., 2012; Erich et al., 2002; Gao et al., 2014; Guppy et al., 2005; Reddy et al., 1980;
Walbridge et al., 1991). On a molecular basis, this can be attributed to the competitive saturation of sorption sties by OC anions as reported in other studies (Antelo et al., 2007; Hunt et al., 2007; Ohno and Crannell, 1996; Staunton and Leprince, 1996) (Figure 7A). Guppy et al. (2005) claim that the observed decrease in Pi sorption is not attributable to competition for sorption sites, but instead to the increased mineralization of P-bearing OC compounds, akin to the White et al. 2014 model (Figure 7B). These mechanisms are not necessarily mutually exclusive, for example, as soils move closer to C saturation, N mineralization
increases because the soils lack new sites for OC to be stabilized on minerals (Castellano et al., 2012; White et al., 2014). Regardless of the proposed mechanisms, in order to improve P models, accounting for competitive sorption reactions (Regelink et al., 2015) or the increase in Pi content from mineralization or Pi release (Guppy et al., 2005) is a necessity.

### 3.3 The contribution of Po to DPS and the interactions between Pi and Po

A common method of characterizing sorption dynamics is to develop a soil sorption isotherm, or sorption curve, which requires
equilibrating various concentrations of Pi with a soil sample. Although to a lesser extent, Po sorption dynamics have also been studied using sorption isotherms. Instead of using Pi in the equilibrating solution, different forms of Po can be added together or





separately with Pi, and their sorption dynamics compared (Anderson et al., 1974; Berg and Joern, 2006; McKercher and Anderson, 1989). The $PO_4$ ion on Po partially controls its sorption so Po compounds share similar sorption dynamics with Pi. Po's affinity to the soil phase increases with the number of phosphate groups (Shang et al., 1990). Due to the steric hindrance of

the C moiety (Celi et al., 1999), it is suspected that Po does not readily penetrate mineral pores (Shang et al., 1992). However, the similar surface binding mechanisms of Po and Pi likely means they compete for surface sorption sites, which has implications for the solubility of both Po and Pi and thus its availability for plant uptake or hydrologic losses.

When multiple Po species and Pi are added to the equilibration solution of a sorption isotherm experiment, the affinity of each species can be measured in competition with one another. Po generally exhibits less efficient sorption than Pi, except for phytate.

Phytate sorption dynamics are debated, with some reports stating that phytate sorbs to a greater extent than Pi (Berg and Joern, 2006; McKercher and Anderson, 1989; Wang et al., 2007) and others showing the opposite result (Lilienfein et al., 2004; Ognalaga et al., 1994; Shang et al., 1990).

An explanation parsimonious with both results is that Po with multiple phosphate groups may sorb more strongly while Pi may sorb more rapidly. The Po sorption strength increases with the number of phosphate groups (Berg and Joern, 2006; Shang et al.,

1990). When compounds like phytate are present in a soil, stabilized by mineral interactions and not readily mineralized, Pi saturation should decrease due to the lower net desorption of Po (Table 1, C1). In contrast, cases showing preferential sorption of Pi over Po might be explained by the lower activation energy needed for a $PO_4$ ion to bind to soil compared to a more complex mono or diester organic compound. Both processes are likely co-occurring; Po may sorb stronger, particularly if it has more $PO_4$ groups, but Pi may sorb faster due to its low activation energy and lower steric hindrance (Shang et al., 1990). Yet, we lack an

understanding of the conditions that cause one of these processes to dominate in a soil environment.

Po saturation may influence Po mineralization dynamics and should be the subject of future empirical and modeling work. This can be critical to understand P pollution from soils with high organic C and high Po, a condition that is becoming more prevalent with the advance of no-till in manured soils or in any soil receiving ample supply of manure.

### 3.4 Synthesis

Advances in understanding Po dynamics and its relation to Pi and OC saturation are required in three areas. First, the contributing role of Po to P saturation and thus to greater P sorption and cycling dynamics must be further investigated. This includes determining the effect of Po on DPS measurements, for Po is implicitly considered in the denominator but not always in the numerator of the DPS calculation. Second, the controls on P sorption of litter application, rhizosphere exudation/deposition, or native OC needs to be clarified. Organic additions can increase Pi in the solution due to Po mineralization (as proposed by

Guppy et al., 2005) or by increasing the solubility of Pi (as proposed by Hunt et al., 2007; Oburger et al., 2011; Regelink et al., 2015). Both processes may be co-occurring but it is critical to define the conditions under which one or the other dominates. And lastly, modeling and experimental efforts are needed to determine the effect of C saturation on Po mineralization, as has been done for N (White et al. 2014). These efforts may need to consider phytate and non-phytate compounds as two separate pools. In soils that are amended with manure, C saturation and P saturation can be critical controls of both C and P turnover.





## 4 Methodological discrepancies between P analyses and implied P dynamics

### 4.1 Soil Pi and Po pools

Soil Pi and Po forms vary in turnover rates along a continuum of stabilization (Tiessen and Moir, 1993). To model their dynamics, Pi and Po are subdivided into pools, with the exception of ECOSYS that simulates Pi dynamics using dozens of element and mineral-specific equilibrium reactions (Grant and Heaney, 1997). In most models these pools are conceptual and are not representable as analytical fractions (Six et al., 2002), such as those determined by the Hedley Fractionation Method. This is in part because analytical fractions do not always respond in a quantifiable manner directly to inputs and outputs (Delve et al., 2009). Despite this limitation, conceptual pools are still estimated by establishing relationships between P lability and P transformations (Delve et al., 2009; Gijsman et al., 1996; Hedley et al., 1982; Stewart and Tiessen, 1987; Tiessen et al., 1984).

Among the consequences of the lack of correspondence between modeled and measured pools are that these models may need calibration and that as Pi and Po decrease or increase there is no assurance that the calibrated parameters values will remain valid outside the calibration domain. P is distributed into pools based on its stability and cycling rate and the more stable pools are thought to be available over decadal timescales (Richter et al., 2006) through the slow replenishment of labile fractions (Guo et al., 2000). However, both inorganic and organic P forms are likely accessible on shorter, seasonal timescales. For example, phytate which is often considered highly stable, has been observed to mineralize rapidly (Doolette et al., 2010). For Pi, extracted pools that are considered "stable", such as the HCl fraction, have a greater short-term turnover than previously thought (Joshi et al., 2016; Siebers et al., 2018), a fact that must be incorporated into models that have invariable cycling rates of stable fractions.

### 4.2 Sorption Curves

Inherently linked to the stabilization of phosphate in various soil phases are the processes and rates of phosphate sorption. Sorption curves, often referred to as sorption isotherms, are used to determine the extent of rapid P sorption between the solution P and the soil (McGechan and Lewis, 2002). Three problems arise from the use of sorption curves: they are soil-dependent rather than generalizable, the methods involved in generating the curve can influence the sorption isotherm parameterization and the modeled P dynamics, and the P sorption-desorption pathways is hysteretic (Barrow, 1983, 2008; Grant and Heaney, 1997; Okajima et al., 1983).

Because sorption isotherms are time-consuming and are not a part of routine soil analyses, P is often modeled using sorption curve parameters that do not always represent the simulated soil. The general applicability of sorption curves is further reduced because they are not mechanistic and do not account for site-specific differences in pH, ionic strength, or surface complexation (Grant et al. 1997; Arai and sparks 2007).

Typically, the generation of sorption curves involves shaking P solutions with a range of known concentrations with a soil sample. After 16-24 hours, the equilibrium P concentration is measured, and the curve is fitted. If the shaking is too vigorous, new sorption sites can be exposed, and the isotherm is no longer representative of P sorption (Barrow 2008). Furthermore, parameterizing the hysteretic nature of P involves time consuming methodologies (Limousin et al., 2007), but its exclusion results in model uncertainty.



Fitting curves generated by isotherm experiments can also be problematic. The equations by Langmuir (Table 2, Eq. 2) and

Freundlich (Table 2, Eq. 3) are commonly used to fit sorption curves. The Langmuir isotherm assumes a reaction with a uniform

sorption surface, which does not occur in soils. Freundlich isotherms assume an infinite number of sorption sites (Kruse et al.,

2015), i.e. no sorption maximum is considered like in the Langmuir. Neither the general Langmuir nor Freundlich equations

account for multiple possible sorption pathways, which may result in their underestimation of sorption (Hussain et al., 2012) and

reduced fit (McGechan and Lewis 2002). Two phase Langmuir equations (Table 2, Eq. 4) can be used, as they have provided

better fit for sorption isotherms in some cases (Holford et al., 1974; Hussain et al., 2012). However, using the same soils as

Holford et al. (1974), Barrow (1978) demonstrated the Freundlich had comparable fits with reduced parameters, which may

explain the absence of two phase Langmuirs in simulation models.

Sharpley et al. (1984) employed a longer and more intensive approach to estimating a "P sorption parameter" (PSP), which is the

percentage of added P remaining in the soil solution after equilibrium is reached with a labile "active" pool (Vadas et al. 2013).

In lieu of a 24-hour incubation, known P concentrations were added to subsamples that were dried and rewetted three times with

DI over a period of six months. Three general equations were developed for calcareous, slightly weathered, and highly weathered

soils. These equations were integrated into the SWAT model (Arnold et al., 1998). Although this model accounts for temperature

and moisture factors, pH is grouped into large categories which may limit its applicability to the highly variable nature of soils.

Furthermore, equation 7 shows that OC has a negative relationship with PSP, which is the opposite of the expected response as

outlined in the C and P saturation section. In SWAT, this PSP variable and the solution P are used to calculate the size of pools

in the model. However, solution P is assumed to be half of Mehlich-3 or other STP. This may work for initialization before

model spin up, or at least provide a consistent method, but is not a representative method in determining P pool sizes.

An alternative to both isotherm methods is the use of known equilibrium constants of various soil minerals and precipitates to

calculate an equilibrium of sorbed or mineral P with the soil. Such an approach was taken by Grant et al. 1997 in the ECOSYS

model. Although based on first principles, it relies on obtaining concentrations of many soil minerals, which may be impractical

or difficult. For reliable parameterization, it would also require knowing the effective mineral-solution contact; larger minerals or

colloids would have a lower surface area for P exchange compared to smaller colloids.

### 4.3 Synthesis

Translating analytical P pools into models is complicated by the fact that extracted pools are not compound-specific and only

provide an estimation of Pi and Po turnover rates. This issue is exacerbated by the diversity among and within chemical

extraction protocols. The most used extraction method, the Hedley Fractionation method, is also the most heavily modified,

having dozens of variations. This is a weak foundation for the comparison and meta-analysis of data from studies employing

variations of the Hedley protocol. If we are to incorporate measurable pools into models, these pools need to be measured with a

consistent protocol. Although efforts in this space have been made (Joshi et al. 2018), it is necessary to continue coupling [31]P

spectroscopy and chemical extractions to determine pool compositions, as well as tracing to assess compound-specific lability.

Conducting sorption isotherms is tedious and is not a part of standard soil tests. Applying isotherm parameters, such as the

binding energy constant ($K_L$), when predicting or generating data can result in poor validation, as the conditions from which the

parameter was derived likely do not match the simulated conditions. If an isotherm approach is to be used, then it will be

necessary to employ methods to rapidly estimate the sorption parameters from accessible and cost-effective techniques, such as

in Dari et al. (2015). In future method development, factors that influence sorption, such as the OC content of the soil, should be



considered. This is of particular importance as P isotherms are sometimes used in determining the saturation capacity, a necessary component of DPS measurements. As previously discussed in the Organic P and C Saturation section, the presence of Po and OC may influence the interpretation of DPS data.

Improved accessibility of mineral analysis methods will facilitate the application of a bottom-up, first principles approach such as the dynamic equilibrium approach of ECOSYS (Grant and Heaney 1997). If the approach of Sharpley et al. 1984 is to be revised and expanded, soil-specific properties (OC, pH, clay content, etc.) must be accounted for in a continuum that can be generalized. Furthermore, pool-pool transfer functions could benefit from the application of isotope tracing techniques; this is an area which clearly requires more fundamental research.

**Conclusions**

Limitations in simulating Pi sorption, Po mineralization, stoichiometry as it relates to P, and nutrient interactions, calls for an improved experimental and modeling framework to interpret P cycling. Our current knowledge of soil P dynamics suffices for the management of crop nutrition, but still lags C and N in terms of process-based modeling (Beegle, 2005). We recommend focusing specifically on 1) updating the McGill and Cole model; 2) understanding the dynamics of phytate and soil stoichiometry; 3) exploring the role of OC and Po saturation on P sorption and Po mineralization, and 4) resolving discrepancies
between hypotheses about P cycling and methods to measure P pools and fluxes.

It is necessary to better understand P processes to build the most representative P simulation models. Understanding the controls on plant, microbial biomass, and microbial stoichiometry will help determine bottom up controls on the ecosystem, biome, and even global P cycling. The challenges summarized in this review will hopefully steer research efforts towards a better empirical understanding of the P cycle that will ultimately reduce the reliance on intensive calibration (Beven et al., 2006). A better
understanding of P cycling will lay the stratum upon which better predictive models will be developed, enabling foresight of both plant P nutrition and P pollution challenges, rather than reacting to them when symptoms like surface water eutrophication manifest visibly and in full force.

**Author Contribution**

ARK, CM, and JPK conceived the manuscript; CM conducted the literature search, organized and summarized information; CM,
ARK and JPK defined the organizational structure after the literature search, CM drafted the manuscript and prepared the figures; ARK and JPK contributed to the discussion and edited the manuscript. ARK and JPK secured funding to support this research.

**Competing Interests**

The authors declare that they have no conflict of interest.

**Acknowledgements**

Funding for this research was provided by USDA National Institute of Food and Agriculture award #2016-67003-24966 and by Hatch Appropriations under Project #PEN04571 and Accession #1003346.





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



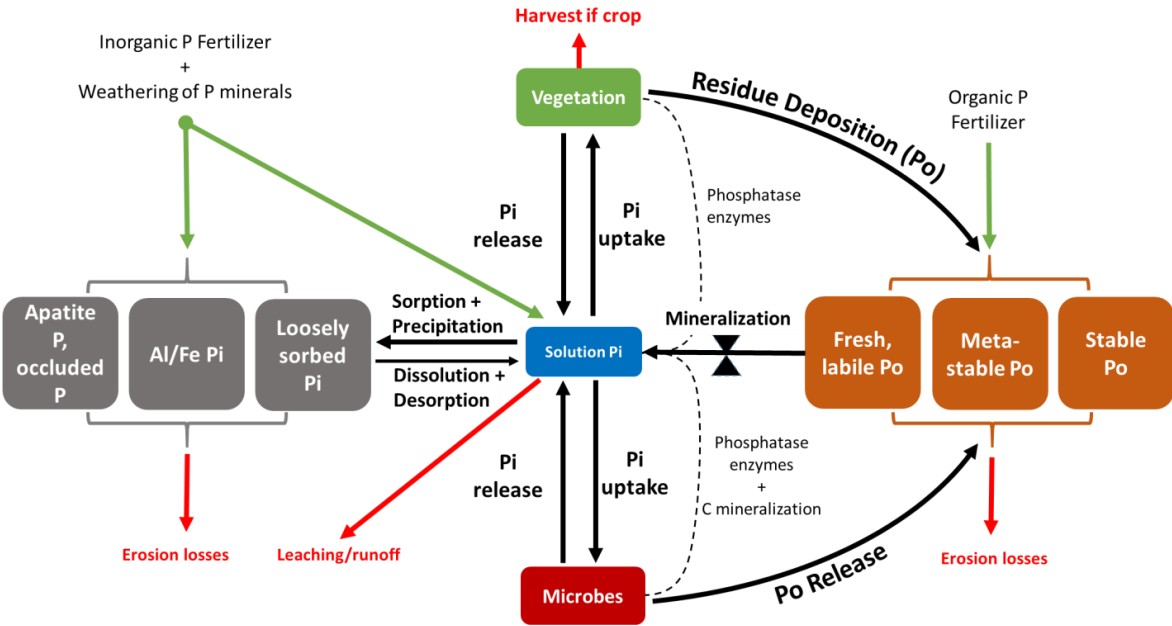

**Figure 1. Conceptual diagram of the soil P cycle. Pi and Po represent inorganic (gray) and organic (orange) phosphates, respectively. The Pi and Po pools that are situated farthest from the depicted solution pool in this diagram are considered more stable. There are small amounts of organic P in solution that are not explicitly depicted here. The bowtie represents the control mineralization exerts on the transformation of P from Po pools to the solution pool. The dotted lines represent enzymes in the mineralization of Po. There are other abiotic controls of mineralization excluded from this diagram.**




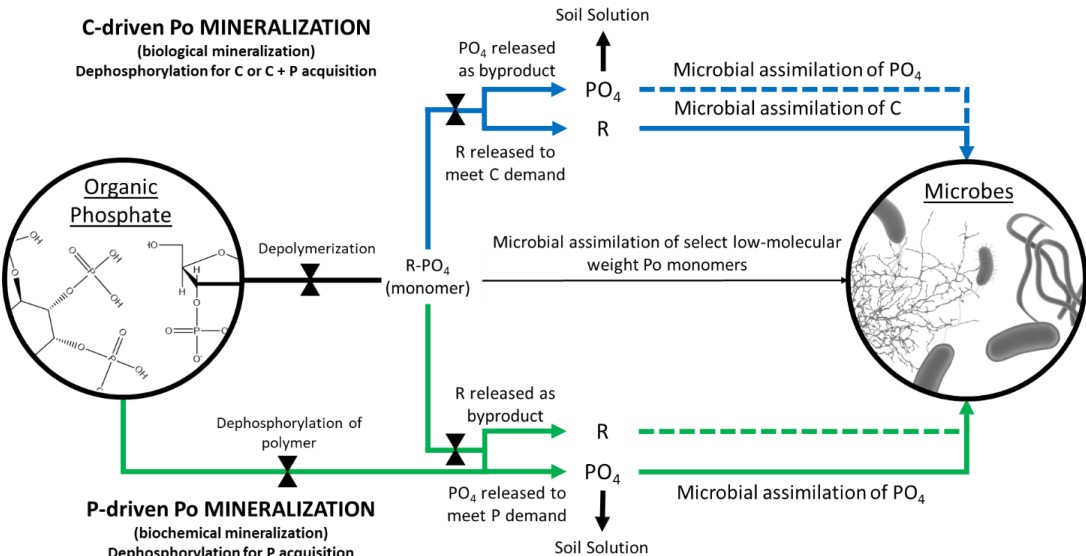

**Figure 2. Conceptual diagram of C and P-driven Po mineralization. Because microorganisms can only directly uptake specific types of orthophosphate ester (Po) monomers directly (thin middle black arrow), the accessibility of Po is controlled by the action of phosphatase enzymes (indicated by bowties). The blue arrows represent C-driven Po mineralization, whereby the organic compound is mineralized for C acquisition, and depending on P demand, for P acquisition (immobilization, indicated by dotted blue line). P that is unused by microbes is left in the soil solution. The green arrows represent P-driven Po mineralization, whereby PO$_4$ is hydrolyzed from the Po compound and assimilated without further C processing (unless the C is available for uptake and is needed, and is circumstantially assimilated, indicated by green dotted line).**

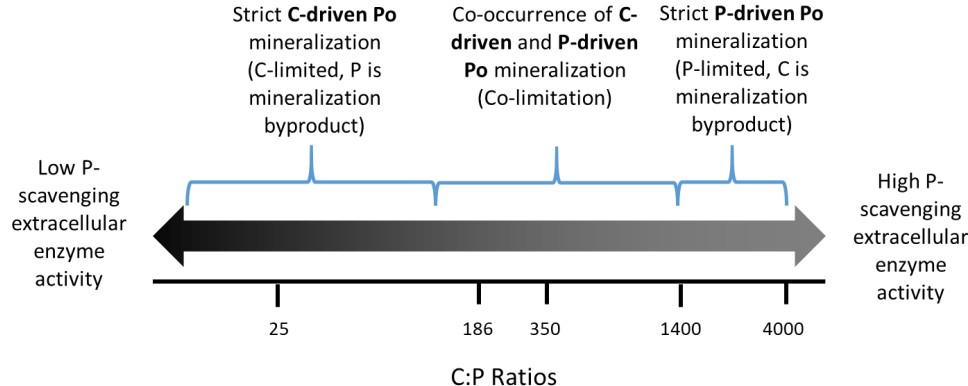

**Figure 3. Hypothesis of stoichiometric controls on C-driven and P-driven Po mineralization. The log-scaled numbers are C:P ratios taken from the literature that include stoichiometry of microbes, litter, soil, and critical ratios. The brackets represent where biological or biochemical mineralization may begin to dominate.**








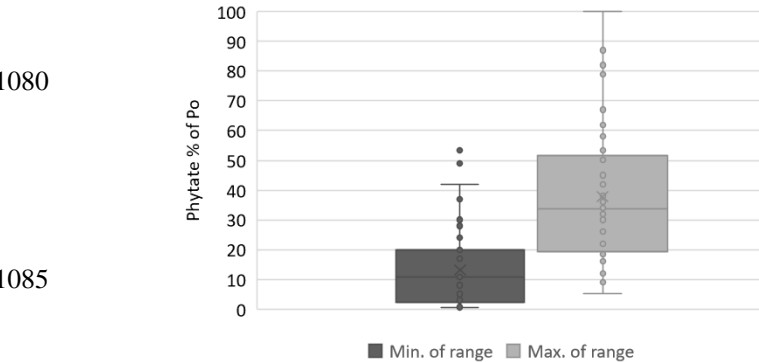


**Figure 4. Minimum and maximum % of Po comprised of phytate observed in a variety of natural and agricultural systems (n=41). Sources: Giles 2014; Turner 2002; Turner 2007.**




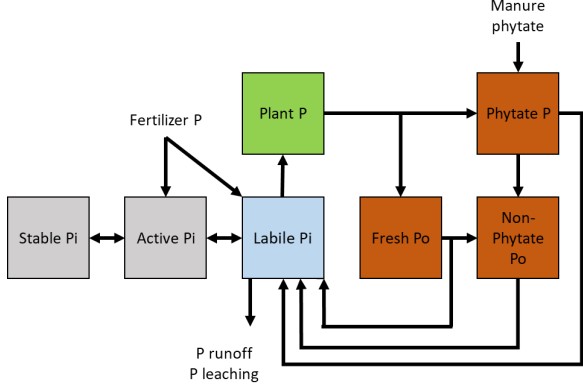

**Figure 5: Simple conceptual diagram for modeling inorganic and organic phosphorus cycling, which includes an explicit phytate pool.**





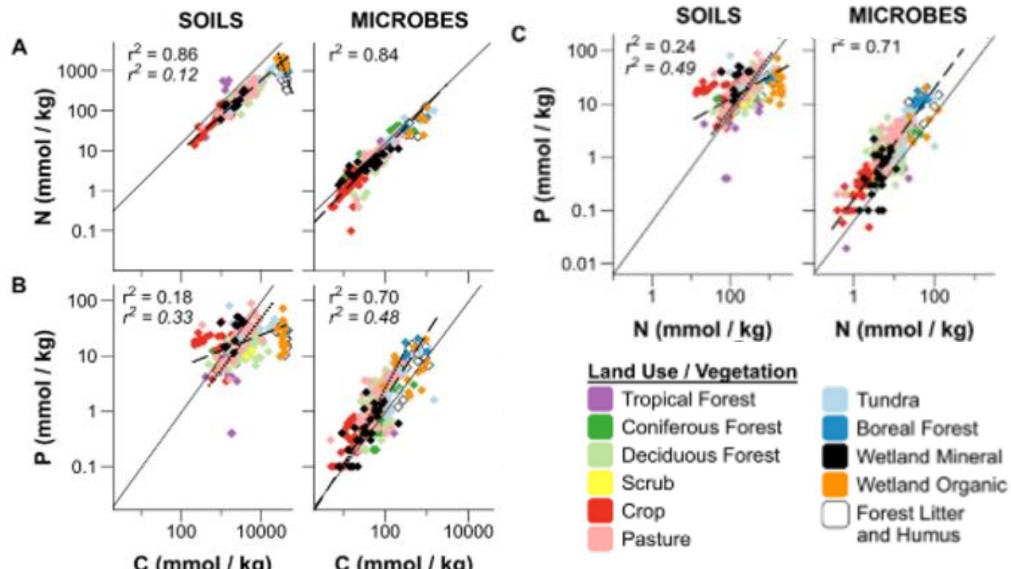

**Figure 6. Regressions of C:N (A), C:P (B), and N:P (C) ratios in soils and the microbial biomass using a $log_{10}$ transformation. The dashed lines (standard font $r^2$) represent the stoichiometric regressions for global soil or microbial biomass measurements, while the solid lines represent the Redfield (1958) ratio. Dotted lines are regressions (italicized $r^2$) for specific habitats whose soil stoichiometry differed significantly from the others. Modified from Hartman and Richardson 2013.**


Table 1. Hypothetical scenarios comparing the measured Pi saturation (DPS) and actual total P saturation.

| Figure | Scenario | Presence of OC | Total Pi (mg P/kg) | Total Po (mg P/kg) | Pi and Po saturaton cap. (mg P/kg) | Measured Pi saturation [DPS] (%) | Po saturation (%) | Actual total P saturation (%) |
|---|---|---|---|---|---|---|---|---|
| N/A | **Fe/Al Oxide Mineral** (No Pi, Po) | No | 0 | 0 | 500 | 0 | 0 | 0 |
| N/A | **Complete Pi saturation** (No Po) | No | 500 | 0 | 500 | 100 | 0 | 100 |
| 7A | **Incomplete Pi saturation** (No Po. OC competing with Pi for sorption) | Yes | 400 | 0 | 500 | 80 | 0 | 100 |
| 7C | **Incomplete Pi saturation** (Po competing with Pi for sorption) | No | 300 | 200 | 500 | 60 | 40 | 100 |

**Actual P saturation is defined as the total saturation of both Po and Pi species. The competition with non-P-bearing OC anions (7A) would lower the measured DPS despite the saturation of all available sorption sites. When Po successfully competes with Pi for sorption (7C), the resulting measured DPS will be lower than the actual P saturation. It is important to note that, as opposed to colorimetric methods, if inductively coupled plasma optical emission spectrometry (ICP-OES) is used in DPS analysis, any circumstantially extracted Po would be included in the measurement, resulting in a higher % DPS than presented here.**


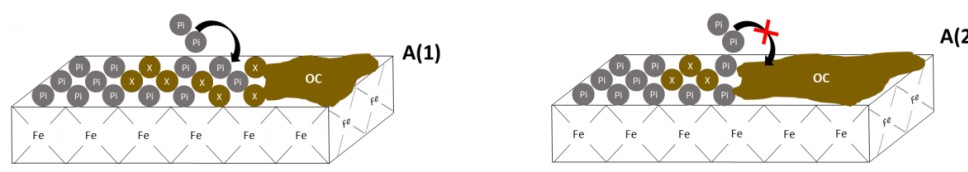

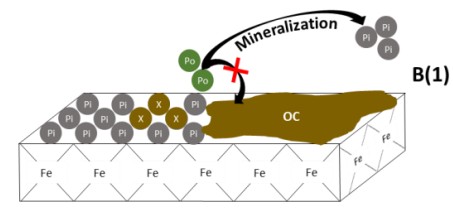

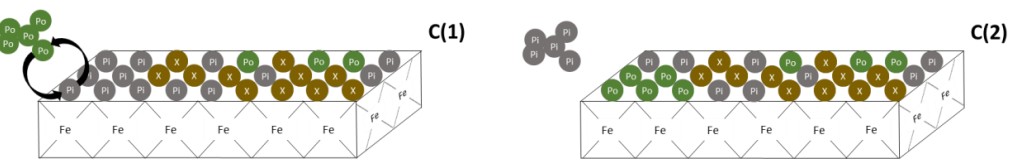

**Figure 7. Conceptual depiction of Pi and Po saturation dynamics. The green Po bubbles are organic phosphate, the gray Pi bubbles are inorganic phosphate, and the brown "X" bubbles are non-P bearing organic and inorganic compounds. (A) represents the competitive sorption of non-P-bearing organic carbon (OC) compounds on the sorption of Pi. As OC saturation increases (A2), so does the solution concentration of Pi. (B) shows the increase in OC from A2 causing an increase in Po mineralization rather than a decrease in Pi sorption. (C) depicts the replacement of Pi with Po on an iron oxide mineral surface, resulting in a change in composition of the saturated surface. Because DPS measurements do not target Po, the perceived DPS would be lower for mineral C2, despite having the same overall P content. The replacement of Pi with Po on sorption surfaces would increase the C:P ratio of the soil.**








Table 2: Equations

| # | Name | Equation | Parameters |
|---|------|----------|------------|
| 1 | C sorption capacity | $C_x = 21.1 + 37.5 \times f_{clay}$ | $C_x$: maximum sorption C capacity (g C kg⁻¹) soil<br>$f_{clay}$: clay concentration (g kg⁻¹) |
| 2 | Langmuir | $\dfrac{C}{S} = \dfrac{C}{S_{max}} + \dfrac{1}{K_L \times S_{max}}$ | **S**: sorbed P (mg kg⁻¹)<br>**C**: equilibrium P concentration (mg L⁻¹)<br>**S**max: sorption maximum (mg kg⁻¹)<br>**K**L: binding energy constant |
| 3 | Freundlich | $S = a \times c^b$ | **S**: sorbed P (mg kg⁻¹)<br>**a**: proportionality constant (mg kg⁻¹/ mg L⁻¹)<br>**b**: empirical binding energy coefficient (dimensionless)<br>**c**: P concentration in solution (mg L⁻¹) |
| 4 | Two-surface Langmuir | $\dfrac{x}{m} = \dfrac{b_I K_I EPC}{1 + K_I EPC} + \dfrac{b_{II} K_{II} EPC}{1 + K_{II} EPC}$ | **x/m**: sorbed P (mg kg⁻¹)<br>**K**I/**K**II: bonding energy, regions I and II (L mg⁻¹)<br>**b**I/**b**II: adsorption maximum, regions I and II (mg kg⁻¹ soil)<br>**EPC**: equilibrium P concentration (mg L⁻¹) |
| 5 | PSP: Calcareous soils | $PSP = 0.58 - 0.0061 * CaCO_3$ | **PSP**: P sorption parameter[*]<br>**CaCO₃**: calcium carbonate (%) |
| 6 | PSP: Slightly weathered soils | $PSP = -.70 - 0.0043 * BS + 0.0034 * Pi_l + 0.11 * pH$ | **BS**: base saturation (%)<br>**P**il: labile P (µg g⁻¹ soil)<br>pH |
| 7 | PSP: Highly weathered soils | $PSP = 0.038 - 0.047 * \ln\left(\dfrac{f_{clay}}{10}\right) + 0.045 * P_{il} - 0.053 * OC$ | $f_{clay}$: clay concentration (g kg⁻¹)<br>**OC**: organic carbon (%) |

Equation 1: Hassink and Whitmore 1997

Equation 2, 3: Barrow 2008, Dari et al. 2015

Equation 4: Zhang et al. 2005

Equation 5-7: Sharpley et al. 1984; Vadas et al. 2010

[*] Also known as "phosphorus availability index" (PAI)