# Peer review of "Reviews and Syntheses: Ironing Out Wrinkles in the Soil Phosphorus Cycling Paradigm"

_Biogeosciences, 2020_

## Referee Comment (RC1) · Anonymous Referee #1 · 1 Jun 2020

I find the paper by McConnell a very interesting piece of work, which coincidently covers lots of our accomplished and undergoing research regarding soil phosphorus cycling. Therefore I would like to discuss with the authors about some interesting points in the paper.

As a modeler working with P processes for years, I totally agree with the four points the authors brought up, and I am impressed by the range of literature they reviewed. However, I am not certain if anyone who does not have enough knowledge about models will be easily convinced by the authors. My overall perception of the paper is that it has a very nice flow of information regarding phosphorus process understandings and techniques, but it lacks in-depth insights of current "P models" to justify its statement that these processes should be improved in models. It is partly because some of

the processes or features are simply ignored in current models, such as phytate, and partly because the proposed processes by authors are not particularly discussed by modeling community yet. Nevertheless, I think the strength of the paper will be much enhanced if the discussion on the modeling side is improved.

First of all, the basis for discussion used by the authors is a very unspecific term, "P cycling models", which in practice does not really count as a sub-category of models. As I see from the few models that the authors cited, SWAT, CENTURY, and ECOSYS are terrestrial ecosystem models focusing on catchment-scale agriculture system, grass/forest/crop, and soil chemistry processes, respectively; CLM-CNP and CASA-CNP are global land surface models, which are developed from site-level terrestrial biosphere models (TBMs). There are a lot of other models fit within these two categories, such as EPIC (Jones et al. 1984), GLEAMS (Leonard et al. 1987), AN-IMO (Schoumans & Groenendijk 2000), and a number of other mathematic models (e.g. Buendía et al. 2010, Runyan &D'Odorico 2012) for the former one; as for the latter one, TBMs that already implement P cycling processes include JSBACH (Goll et al. 2012), ELM (Zhu et al. 2016), ORCHIDEE (Goll et al. 2017), ForSAFE (Yu et al. 2018), QUINCY (Thum et al. 2019). I noticed that the authors also try to cover OC processes in the paper, and there is also many soil organic carbon models, such as MIMICS (Wieder et al. 2014), MEND (Wang et al. 2015), RESOM(Tang and Riley 2015), COMISSION (Arhens et al. 2015), and JSM (Yu et al. 2020). I understand it might be too ambitious to cover all details in process understandings, techniques, and model descriptions/limitations, but perhaps the authors could try to narrow down the scope of models, for example, only to land surface models and TBMs.

Secondly, from a modeler point of view, the four points could be organized in a way that is easier to follow. The most intriguing points for me (modelers) are those processes/features that are already described in models and need better understandings to improve their model descriptions. These processes/features are the biochemical Po mineralization, the stoichiometry problem, and the methodological discrepancies

between P analyses and implied P dynamics. The other processes/features are also interesting but much less noticed by the modeling community. For example, I think the connection between stoichiometry and C- and P-driven mineralization is much stronger than phytate.

Below are some detailed comments following the order of sections.

Section 1:

Line 118: the modeling work has also supported that CP cycling is largely decoupled from CN cycling (Yu et al. 2020, GMD)

Line 134: Is the sentence complete?

Line 164: logically, I think it is better to place the last sentence at the beginning of this paragraph.

Line 187: Schimel and Weintraub did not simulate phosphatase

Line 194: please check Yu et al. 2020, in which a dynamic enzyme allocation approach is used to mimic such a relationship

Line 209: I would argue many terrestrial ecosystem models (ESMs) did include the P-driven mineralization. From as early as the CENTURY model (Parton et al. 1988) to the more recent ESMs such as JSBACH (Goll et al. 2012), ORCHIDEE (Goll et al. 2017), ForSAFE (Yu et al. 2018), E3SM (Zhu et al. 2019), QUINCY (Thum et al. 2019). And the overall role of P-driven mineralization (biochemical mineralization) is crucial for plant growth in some of these models.

Section 2

Line 325: please check Lang et al. 2017, SBB.

Line 366: The real challenge (for modeling) is to account for the huge stoichiometry differences between plant litter, microbe, and SOM, especially C:P ratio (Xu et al., 2013;

Mooshammer et al., 2014). This requires an explicit microbial pool and a number of microbial adaptation processes to be included in models. Additionally, how to extrapolate these mechanisms from site-level to regional and global level is another complicated problem

Section 3

Line 398: there are some models implementing the OC saturation dynamics (or similarly clay-related C sorption capacity), such as MIMICS (Wieder et al. 2014), MEND (Wang et al. 2015), RESOM (Tang and Riley 2015), and COMISSION (Arhens et al. 2015). The recent JSM (Yu et al. 2020) has also include N and P in the OC sorption, which also complies with the saturation principle, although Po is not specifically separated as a competing sorbate of OC

Line 445: in principle, what is discussed in this paragraph makes sense, however, given the existing uncertainties in Pi sorption (as discussed in section 4), it is really a challenge ahead of our current focus to consider the role of Po and interactions with OC saturation. The role of OC in Pi sorption has been partially considered in QUINCY (Thum et al. 2019), which proves to have an important role in regulating the P availability and thus affect plant growth, but this is not specifically discussed in the model description paper.

Section 4

Line 501, 506: DI and STP not defined Line 517: The work by Helfenstein et al. 2020, BG, has shown that there is a certain level of correlation between the Hedley Pi pool and Isotopic Exchange Kinetics Pi pool. And our recent work actually shows, with the implementation of double-surface Langmuir on current Pi pool structure, there is a possibility of utilizing the Hedley data for model validation

Line 525: Dari et al. 2015 not found in reference. And the effect of OC content on Pi sorption is already partially implemented in QUINCY (Thum et al. 2019, GMD)

Figure 1: Isn't the weathering of P coming from apatite P? are the three Pi and Po pools forming a continuum of stability? Do they transfer with each other? If they do, please add internal fluxes between pools. If they don't, do they all directly transfer phosphate to solution Pi?

Figure 3: Are there any references for the C:P ratios values in the figure?

Figure 6: the resolution of the figure is too low to read. What are the differences between the two dash lines in each sub-panel?

Table 2: it needs an appropriate caption. There are quite some different equations for C sorption capacity and PSP, why do you choose those specific ones. I am not particularly sure about the purpose of these equations here, particularly the ones calculating PSP.

Reference: Sharpley, A. N., Jones, C. A., Gray, C. & Cole, C. V. (1984). Soil Science Society of America Journal 48, 805-809.

Leonard, R. a., Knisel, W. G. & Still, D. a. (1987). Transactions of the American Society of Agricultural Engineers 30, 1403-1428.

Schoumans, O. F. & Groenendijk, P. (2000). Journal of Environmental Quality 29, 111-116.

Buendía, C., Kleidon, A. & Porporato, A. (2010). Biogeosciences 7, 2025-2038.

Runyan, C. W. & D'Odorico, P. (2012). Advances in Water Resources 35, 94-109.

Goll, D. S., Brovkin, V., Parida, B. R., Reick, C. H., Kattge, J., Reich, P. B., Van Bodegom, P. M. & Niinemets, Ü. (2012). Biogeosciences 9, 3547-3569.

Goll, D. S., Vuichard, N., Maignan, F., Jornet-Puig, A., Sardans, J., Violette, A., Peng, S., Sun, Y., Kvakic, M., Guimberteau, M., Guenet, B., Zaehle, S., Penuelas, J., Janssens, I. & Ciais, P. (2017). Geosci. Model Dev. 10, 3745-3770.

Thum, T., Caldararu, S., Engel, J., Kern, M., Pallandt, M., Schnur, R., Yu, L. & Zaehle,

S. (2019). Geosci. Model Dev. 12, 4781-4802.

Zhu, Q., Riley, W. J., Tang, J., Collier, N., Hoffman, F. M., Yang, X. & Bisht, G. (2019). Journal of Advances in Modeling Earth Systems 11, 2238-2258.

Yu, L., Zanchi, G., Akselsson, C., Wallander, H. & Belyazid, S. (2018). Ecological Modelling 369, 88-100.

Yu, L., Ahrens, B., Wutzler, T., Schrumpf, M. & Zaehle, S. (2020). Geosci. Model Dev. 13, 783-803.

Wieder, W. R., Grandy, A. S., Kallenbach, C. M. & Bonan, G. B. (2014). Biogeosciences 11, 3899-3917.

Tang, J. & Riley, W. J. (2014). Nature Climate Change 5, 56.

Wang, G., Jagadamma, S., Mayes, M. A., Schadt, C. W., Megan Steinweg, J., Gu, L. & Post, W. M. (2014). The Isme Journal 9, 226.

Xu, X., Thornton, P. E. & Post, W. M. (2013). Global Ecology and Biogeography 22, 737-749.

Mooshammer, M., Wanek, W., Zechmeister-Boltenstern, S. & Richter, A. (2014). Frontiers in Microbiology 5.

Lang, F., Krüger, J., Amelung, W., Willbold, S., Frossard, E., Bünemann, E. K., Bauhus, J., Nitschke, R., Kandeler, E., Marhan, S., Schulz, S., Bergkemper, F., Schloter, M., Luster, J., Guggisberg, F., Kaiser, K., Mikutta, R., Guggenberger, G., Polle, A., Pena, R., Prietzel, J., Rodionov, A., Talkner, U., Meesenburg, H., von Wilpert, K., Hölscher, A., Dietrich, H. P. & Chmara, I. (2017). Biogeochemistry.

---

## Referee Comment (RC2) · Clive Kirkby (Referee) · 8 Jun 2020

Overall I think the review is very good although, as I am not a modeller, I will be commenting more on specific methodological issues.

My own work has shown that while soil C:Pt is extremely variable and highly likely not the parameter we should be looking at C:Po is not as variable and probably much more useful. This of course raises the issue of a reliable and repeatable method for measuring Po which I do not think we have yet. While the authors mention methodological and analytical discrepancies it is done in just a couple of lines. Do the authors think it is really important or not so important, and why. While it is pretty obvious why Pi is generally poorly correlated with soil C do the authors think soil C is poorly correlated

with Po because of the unnamed methodological or analytical discrepancies or some other, perhaps unknown, reason.

The authors suggest that as microbes are a major driver of P transformation, which may be driven by their stoichiometry, then differences or shifts in community composition could affect such transformation. Ignoring archaea, for which there is little published information, fungi and bacteria are therefore the two groups that are doing these transformation. It is generally recognised that fungi are more nutrient poor than bacteria (fungal C:P 300-1190 and bacteria C:P 5-370) and thus trying to relate fungal:bacterial ratios of different soils with P transformations maybe useful. While obtaining such a measurement is probably expecting too much it might be worthwhile seeing if forest soils and top soils in no till agriculture (which generally have high fungi:bacteria ratios) can be modelled differently to soils that are often cultivated (which often have lower fungi:bacteria ratios). Taking this to the extreme one could try modelling forest or top soils (high fungi:bacteria ratios) with soils from deeper in the profile, which often have much lower fungi:bacteria ratios.

The difference in fungal and bacterial C:P ratios may help to explain differences in P-driven Po mineralization compared to C-driven Po mineralization in different soils. As fungi do not require as much P as bacteria perhaps C-driven Po mineralization is more common in fungal dominated soils (forest or no till top soils) as they might be mainly after the energy but P-driven Po mineralization might be more common in soils with lower fungi:bacteria as bacteria might be mainly interested in the P for biosynthesis.

---

## Author Comment (AC1) · 20 Jun 2020

Dear reviewer, Thank you for your incisive review of the MS. We hope to address all your comments here. The references provided are also very pertinent and informative and will improve the manuscript.

R1: "...it lacks in-depth insights of current "P models" to justify its statement that these processes should be improved in models. It is partly because some of the processes or features are simply ignored in current models, such as phytate, and partly because the proposed processes by authors are not particularly discussed by modeling community yet."

Authors: The original intent of the review was to compare individual models by re-

constructing their respective P cycling components. That intent veered to the current manuscript with emphasis on missing components. As Referee 1 (R1) noted, there are many terrestrial biosphere models (TBMs) and landscape/watershed models. It was tedious and not necessarily informative to reconstruct them all due to the many overlapping frameworks and processes.

With knowledge of individual models from this earlier research, we redirected our efforts to address concepts not generally (or never) included (for example, phytate dynamics) or those that may not be representing a specific process completely (P-driven "biochemical" mineralization). We believe it is important for the development of biogeochemical models and experimentation to draw attention to those concepts ignored in models or not yet discussed by the modeling community.

In response to the comment regarding "improving P models", we think that the matter is more semantic than conceptual. We may also say "include processes not addressed in current models", which of course would be done to improve models. Our aims are to highlight areas of knowledge that need to be included in models in relation to P cycling, to acknowledge the need to converge on common or complementary frameworks well supported by theory, and to acknowledge that uncertainty still exists due to conceptual ambiguity or the ability to measure P pools and fluxes.

R1: "...'P cycling models', which in practice does not really count as a sub-category of models"

Authors: The semantics here can be subjective. Any model refers to a system with arbitrary boundaries. Whatever comes from outside the boundaries are "forcings" or inputs. Any model "component", let's call it a submodel if inside a well bounded model, is in itself a model with its own forcings. Maybe we can make a stop here and highlight that is difficult to be a purist: setting boundaries does not make the boundaries correct, as there are feedbacks that could affect the forcing variables themselves. For example, a change in land use will affect the air temperature and air moisture through

changes in the surface energy balance. For models that couple the atmosphere with the earth surface, climate variables are mostly forced in, regardless of the land use, and the feedback is ignored. The errors introduced by ignoring the feedback are in most cases pretty minor (and of course there is no assurance that including a feedback will make the outcome better). In any case, the model and sub-model language is context dependent. We will try to avoid any confusion.

R1: "...perhaps the authors could try to narrow down the scope of models, for example, only to land surface models and TBMs"

Authors: R1 provided many models which we could include as examples of certain P processes. In terms of catchment/watershed-scale models, R1 suggested including EPIC (Jones et al., 1984), GLEAMS (Leonard et al., 1987), and ANIMO (Schoumans and Groenendijk, 2000) in our review. GLEAMS was a standalone model. During the early stages of the development of EPIC, it absorbed GLEAMS to represent P processes. SWAT (as well as APEX) uses EPIC as the base crop and soil model for each simulation unit (SWAT is a semi-distributed watershed model and APEX is a pseudo-distributed model mostly applicable to smaller watersheds) (Gassman et al., 2007; Wang et al., 2012). In using SWAT, we were encompassing this conglomerate and avoiding cumbersome explanations and redundancy. We will henceforth refer to SWAT when dealing specifically with SWAT and to GLEAMS-X when referring to models that used GLEAMS components. We will also include a more in-depth analysis of the models recommended by the referee (JSM, QUINCY, COMISSION, SEAM, MEND, MIMIC, ECA, etc.).

Section 1: R1 in reference to Line 118: "the modeling work has also supported that CP cycling is largely decoupled from CN cycling (Yu et al. 2020, GMD)"

Authors: We had missed the work of Yu et al. (2020) while finishing up the review. Indeed, Yu et al. 2020 is a good example of a model representing CP-CN decoupling. This would be an appropriate source for our review. We agree with other comments in

the review. The uncritical inclusion of a Century-like approach and the need to represent soil properties variation with depth as an emergent, and not an imposed property, has also been pointed out and partially addressed by using the carbon saturation concept (e.g. Kemanian and Stöckle, 2010) and so did Yu et al. (2020). We think that conceptually, Yu et al. (2020) and our review are convergent.

R1 in reference to Line 187: "Schimel and Weintraub did not simulate phosphatase"

Authors: The intention was not to convey that Schimel and Weintraub (2003) used phosphatase activity in their theoretical model, rather to highlight that improving phosphatase activity assays would benefit model development employing approaches like that of Schimel and Weintraub, who account for enzymes in SOM decomposition.

R1 in reference to Line 194: "please check Yu et al. 2020, in which a dynamic enzyme allocation approach is used to mimic such a relationship"

Authors: The point that we were making referred to measuring or explicitly modeling the relative release of phosphatases by both plants and microbes. The Yu et al. (2020) use of SEAM (Wutzler et al., 2017) and the ECA approach (Tang and Riley, 2013) addresses a different if related area, pertaining to the dynamic allocation of microbial phosphatases to litter or microbial residue and nutrient acquisition by plants and microbes.

R1 in reference to Line 209: "I would argue many terrestrial ecosystem models (ESMs) did include the P-driven mineralization. From as early as the CENTURY model (Parton et al. 1988) to the more recent ESMs such as JSBACH (Goll et al., 2012), ORCHIDEE (Goll et al., 2017), For SAFE (Yu et al., 2018), E3SM (Zhu et al., 2019), QUINCY (Thum et al., 2019). And the overall role of P-driven mineralization (biochemical mineralization) is crucial for plant growth in some of these models."

Authors: We agree that many models include P-driven mineralization implicitly (CENTURY) (Achat et al., 2016) but those that do so explicitly (Goll et al., 2012, 2017; Yang

et al., 2014) are parameterized on limited data (Reed et al., 2015). Goll et al. 2012 stated that biochemical mineralization "parameter values for the adjustment intensity is set arbitrarily" and that in-field P-driven (biochemical) mineralization assays are not currently possible, thus limiting data. This is similarly stated by Thum et al. 2019 in reference to P-driven mineralization: "Further observations are required to constrain this flux better (Reed et al., 2011)." In other words, one can easily come up with a P flux from P-driven mineralization given P demand through a sort of inverse modeling. Our argument was that P-driven mineralization is not always explicitly included, and it is difficult to obtain P-driven mineralization as an emergent property. When P-driven mineralization is explicitly modeled, it is not well constrained. We could include more models into that categorization to make things clear and the review more thorough.

Section 2: R1 in reference to Line 325: "please check Lang et al. 2017, SBB."

Authors: The work by Lang et al. 2017 was extensive and informative and we will expand on changing P nutrition with depleting P stocks. Lang et al. 2017 is a very fitting reference for the "potential shift to P-driven Po mineralization processes" statement (Line 324-25). The work by Lang et al. 2017 reflects the Walker Syers model (1976), showing that as soils weather and overall P stocks degrade, Po becomes the more available P source as Pi is either lost from the system or occluded. Although not explicitly stated by Walker and Syers, McGill and Cole (1981) discussed the change in nutrient strategies with the depletion of P stocks: "Although Po has not started to decline, it has leveled off and the decline in total P is substantial. Under such conditions demand for internal cycling of Po to meet the needs for P increases."

The objective of the paragraph in question referenced by R1 was to describe how C:P ratios were variable and that they become more decoupled (wider range of possible C:P ratios) as soils weather. This may be due to the depletion in P stocks and changes in P nutrition strategies. It could also be due to variable C demand and the assumed prevalence of extracellular "P-driven 'biochemical' mineralization" to scavenge P. In a highly weathered soil where P availability is limiting, P could be scavenged without

[Figure]

C mineralization, thus further decoupling C and P. Therefore, this decoupling would be dependent to some degree on the relative P to C demand and the extracellular mineralization of Po.

R1 in reference to Line 366: "The real challenge (for modeling) is to account for the huge stoichiometry differences between plant litter, microbe, and SOM, especially C:P ratio (Xu et al., 2013; Mooshammer et al., 2014). This requires an explicit microbial pool and a number of microbial adaptation processes to be included in models. Additionally, how to extrapolate these mechanisms from site-level to regional and global level is another complicated problem"

Authors: We agree. Our intention was to convey this in the review while acknowledging that phytate dynamics are possibly related to the wide ratios in C and Po stoichiometry. The issues identified by the reviewer are on point but were also well understood. The dynamics of phytate is barely in the modeler's radar. But phytate is clearly just a component of the larger picture. We will rebalance the narrative.

Section 3: R1 in reference to Line 398: "there are some models implementing the OC saturation dynamics (or similarly clay-related C sorption capacity), such as MIMICS (Wieder et al. 2014), MEND (Wang et al. 2015), RESOM (Tang and Riley 2015), and COMISSION (Arhens et al. 2015). The recent JSM (Yu et al. 2020) has also include N and P in the OC sorption, which also complies with the saturation principle, although Po is not specifically separated as a competing sorbate of OC"

Authors: Wielder et al 2014 (MIMICS) include the effect of clay on SOM protection, not the concept of carbon saturation which is different (see Table 1 in that paper, row for Cscalar).

Wang et al 2013 and 2015 (MEND) does not address carbon saturation explicitly. There are parameters to represent C affinity to the mineral matrix, but no explicit mention of soil C saturation. Perhaps it is an emergent property of the model. However, the steady state mineral associated organic matter in equation 25 in Wang et al 2013 (variable M)

does not seem to be saturating (i.e. when inputs go to infinity SOC approaches an asymptote). That paper presents steady state equations, but not the limit when inputs go to infinity.

Tang and Riley (2015), figure 12b in the supplemental shows a linear increase in SOM with increasing mineral surface area (many models surrogate MSA with clay content despite the limitation of such simplification). There is however no indication of saturating behavior, although it might be an emergent property if the model is tested with increasing C inputs. It should also consider aggregation (see work by Six et al team, Kong et al., 2005) Tang and Riley model does not address aggregation.

Ahrens et al. (2015) is explicitly saturating (Table 1, Eq 6 in their paper) and as acknowledged in that paper, it follows work by Hassink and Whitmore (1997).

Ignoring fluxes between layers, the soil carbon steady state in Arhens et al is: $Cq = Kads * qmax /(Kads + kdes/Cdoc)$

Cq is soil organic carbon, qmax is the maximum allowed soc, Cdoc is carbon in DOC, and Kads and kdes are DOC sorption and desorption rates. As Cdoc goes to infinity $Cq = qx$.

This is conceptually and mathematically similar to Hassink and Whitmore (1997) model, Kemanian et al (2006), Kemanian and Stockle (2010), and in a more formal formulation in Kemanian et al. (2011). A paper by White et al. (2014) in Biogeosciences followed up on the impacts on N dynamics. Arhens et al. (2015) is a good reference to include in this review, acknowledging that there are earlier precedents. And there is always a possibility that we are missing other relevant work.

We think that the general point that organic C saturation (not the response to clay) is not explicitly included in many models stands, as well as the need to consider the implications for P cycling, much as White et al. (2014) addressed the implications for N.

R1 in reference to Line 445: "in principle, what is discussed in this paragraph makes sense, however, given the existing uncertainties in Pi sorption (as discussed in section 4), it is really a challenge ahead of our current focus to consider the role of Po and interactions with OC saturation. The role of OC in Pi sorption has been partially considered in QUINCY (Thum et al. 2019), which proves to have an important role in regulating the P availability and thus affect plant growth, but this is not specifically discussed in the model description paper."

Authors: Thank for the comment. It seems that R1 agrees with the point. Stating the knowledge gap does neither ignore ongoing efforts to address it (even if they have not been published yet) nor the difficulties inherent to the problem.

Section 4: R1 in reference to Line 501, 506: "DI and STP not defined"

Authors: DI = dionized water, STP = Soil test phosphorus

R1 in reference to Line 517: "The work by Helfenstein et al. 2020, BG, has shown that there is a certain level of correlation between the Hedley Pi pool and Isotopic Exchange Kinetics Pi pool. And our recent work actually shows, with the implementation of double-surface Langmuir on current Pi pool structure, there is a possibility of utilizing the Hedley data for model validation"

Authors: An excellent point, a noteworthy inclusion in our review. However, the point we were making was that the numerous modifications of the Hedley method complicate the analysis of Hedley data. We did not intend to say that correlations can't be drawn, just that the lack of methodological consistency makes this difficult or contributes to a lingering uncertainty about the compatibility of chemically extracted and modeled P pools.

R1 in reference to Line 525: "Dari et al. 2015 not found in reference. And the effect of OC content on Pi sorption is already partially implemented in QUINCY (Thum et al. 2019, GMD)"

Authors: Dari et al. 2015 reference is listed below. Its exclusion was unintentional.

Dari, B., Nair, V. D., Colee, J., Harris, W. G. and Mylavarapu, R.: Estimation of phosphorus isotherm parameters: a simple and cost-effective procedure, Front. Environ. Sci., 3(October), 1–9, doi:10.3389/fenvs.2015.00070, 2015.

R1 in reference to Figure 1: "Isn't the weathering of P coming from apatite P? are the three Pi and Po pools forming a continuum of stability? Do they transfer with each other? If they do, please add internal fluxes between pools. If they don't, do they all directly transfer phosphate to solution Pi?"

Authors: Yes, apatite P is the most prevalent source, the use of "P minerals" was purposefully generic, and referring to it as apatite seems to be most common in the literature. There is likely exchange between the various pools through stabilization and destabilization as mineral-PO4 bonds are formed and broken. However, there should be direct exchange between all soil pools and the soil solution. This is particularly pertinent to organic phosphates. There may be more "stable" Po forms, such as phytate, but P can be hydrolyzed using specific phosphatases such as phytase. P from any Po pool can be directly hydrolyzed into the soil solution, albeit at different rates depending on the stability of the Po form under consideration.

Actual "pools" are less well-defined than those operationally defined by chemical extractions. But for clarity, we could add arrows indicating internal fluxes between Pi and Po pools. All pools can directly transfer to the solution pool; P does not have to cascade down the gradient of pool stability until it reaches the "solution" pool. It is likely that all pools exchange P with the solution and with one another. Representing this will increase the number of arrows and complexity of diagram, but will make it more representative.

R1 in reference to Figure 3: "Are there any references for the C:P ratios values in the figure?"

Authors: Value (description) Reference. Microbial biomass (MB). Threshold Elemental Ratio (TER) 25 (MB C:P ratio, low of range) Capek et al. 2016; 186 (Soil C:P ratio, global average) Cleveland et al. 2007; 350 (MB C:P ratio, high of range) Capek et al. 2016; 1400 (TER, Heuck et al. 2016; Zechmeister-Boltenstern et al. 2015); 4000 (Decomposition stop point, Zechmeister-Boltenstern et al. 2015)

The references will be added to Figure 3.

R1 in reference to Figure 6: "the resolution of the figure is too low to read. What are the differences between the two dash lines in each sub-panel?"

Authors: We will improve the resolution. The dashed lines are regressions for all the Land Use/Vegetation categories in the study. The dotted lines are regressions for Land Use/Vegetation whose slope was significantly different than other systems (e.g. the "Wetland Organic" slope is negative in the top-left regression). We could describe which system regressions differ from the rest in the figure legend if that improves clarity. The important point is that the r2 values are lower for C:P in soils and microbes than the C:N.

R1 in reference to Table 2: "it needs an appropriate caption. There are quite some different equations for C sorption capacity and PSP, why do you choose those specific ones. I am not particularly sure about the purpose of these equations here, particularly the ones calculating PSP."

Authors: PSP equations are listed because they were mentioned in Line 498. The work can be solely cited and equations excluded if that would suffice. But showing the equations makes the model structure unequivocal.

References Achat, D. L., Augusto, L., Gallet-Budynek, A. and Loustau, D.: Future challenges in coupled C–N–P cycle models for terrestrial ecosystems under global change: a review, Biogeochemistry, 131(1–2), 173–202, doi:10.1007/s10533-016-0274-9, 2016.

[Figure]

Ahrens, B., Braakhekke, M. C., Guggenberger, G., Schrumpf, M. and Reichstein, M.: Contribution of sorption, DOC transport and microbial interactions to the 14C age of a soil organic carbon profile: Insights from a calibrated process model, Soil Biol. Biochem., 88, 390–402, doi:10.1016/j.soilbio.2015.06.008, 2015.

Dari, B., Nair, V. D., Colee, J., Harris, W. G. and Mylavarapu, R.: Estimation of phosphorus isotherm parameters: a simple and cost-effective procedure, Front. Environ. Sci., 3(October), 1–9, doi:10.3389/fenvs.2015.00070, 2015.

Gassman, P. W., Reyes, M. R., Green, C. H. and Arnold, J. G.: The Soil and Water Assessment Tool: Historical Development, Applications, and Future Research Directions, Trans. ASABE, 50(4), 1211–1250, doi:10.13031/2013.23637, 2007.

Goll, D. S., Brovkin, V., Parida, B. R., Reick, C. H., Kattge, J., Reich, P. B., Van Bodegom, P. M. and Niinemets, Ü.: Nutrient limitation reduces land carbon uptake in simulations with a model of combined carbon, nitrogen and phosphorus cycling, Biogeosciences, 9(9), 3547–3569, doi:10.5194/bg-9-3547-2012, 2012.

Goll, D. S., Vuichard, N., Maignan, F., Jornet-Puig, A., Sardans, J., Violette, A., Peng, S., Sun, Y., Kvakic, M., Guimberteau, M., Guenet, B., Zaehle, S., Penuelas, J., Janssens, I. and Ciais, P.: A representation of the phosphorus cycle for ORCHIDEE (revision 4520), Geosci. Model Dev., 10(10), 3745–3770, doi:10.5194/gmd-10-3745-2017, 2017.

Hassink, J. and Whitmore, A. P.: A Model of the Physical Protection of Organic Matter in Soils, Soil Sci. Soc. Am. J., 61(1), 131–139, doi:10.2136/sssaj1997.03615995006100010020x, 1997.

Helfenstein, J., Pistocchi, C., Oberson, A., Tamburini, F., Goll, D. S. and Frossard, E.: Estimates of mean residence times of phosphorus in commonly considered inorganic soil phosphorus pools, Biogeosciences, 17(2), 441–454, doi:10.5194/bg-17-441-2020, 2020.
Interactive
comment

Jones, C. A., Cole, C. V., Sharpley, A. N. and Williams, J. R.: A Simplified Soil and Plant Phosphours Model: I. Documentation, , 2, 800–805, 1984.

Kemanian, A. R. and Stöckle, C. O.: C-Farm: A simple model to evaluate the carbon balance of soil profiles, Eur. J. Agron., 32(1), 22–29, doi:10.1016/j.eja.2009.08.003, 2010.

Kemanian, A. R., Julich, S., Manoranjan, V. S. and Arnold, J. R.: Integrating soil carbon cycling with that of nitrogen and phosphorus in the watershed model SWAT: Theory and model testing, Ecol. Modell., 222(12), 1913–1921, doi:10.1016/j.ecolmodel.2011.03.017, 2011.

Kong, A. Y. Y., Six, J., Bryant, D. C., Denison, R. F. and van Kessel, C.: The Relationship between Carbon Input, Aggregation, and Soil Organic Carbon Stabilization in Sustainable Cropping Systems, Soil Sci. Soc. Am. J., 69(4), 1078–1085, doi:10.2136/sssaj2004.0215, 2005.

Leonard, R. A., Knisel, W. G. and Still, D. A.: GLEAMS: Groundwater Loading Effects of Agricultural Management Systems, Trans. ASAE, 30(5), 1403–1418, doi:10.13031/2013.30578, 1987.

Reed, S. C., Townsend, A. R., Taylor, P. G. and Cleveland, C. C.: Phosphorus Cycling in Tropical Forests Growing on Highly Weathered Soils, edited by E. Bünemann, A. Oberson, and E. Frossard, Springer Berlin Heidelberg, Berlin, Heidelberg., 2011.

Reed, S. C., Yang, X. and Thornton, P. E.: Incorporating phosphorus cycling into global modeling efforts: A worthwhile, tractable endeavor, New Phytol., 208(2), 324–329, doi:10.1111/nph.13521, 2015.

Schimel, J. P. and Weintraub, M. N.: The implications of exoenzyme activity on microbial carbon and nitrogen limitation in soil: A theoretical model, Soil Biol. Biochem., 35(4), 549–563, doi:10.1016/S0038-0717(03)00015-4, 2003.

Schoumans, O. F. and Groenendijk, P.: Modeling soil phosphorus levels and phos-

phorus leaching from agricultural land in the Netherlands, J. Environ. Qual., 29(1), 111–116, doi:10.2134/jeq2000.00472425002900010014x, 2000.

Tang, J. Y. and Riley, W. J.: A total quasi-steady-state formulation of substrate uptake kinetics in complex networks and an example application to microbial litter decomposition, Biogeosciences Discuss., 10(6), 10615–10683, doi:10.5194/bgd-10-10615-2013, 2013.

Thum, T., Caldararu, S., Engel, J., Kern, M., Pallandt, M., Schnur, R., Yu, L. and Zaehle, S.: A new terrestrial biosphere model with coupled carbon, nitrogen, and phosphorus cycles (QUINCY v1.0; revision 1772), Geosci. Model Dev. Discuss., (C), 1–38, doi:10.5194/gmd-2019-49, 2019.

Wang, X., Williams, J. R., Gassman, P. W., Baffaut, C., Izaurralde, R. C., Jeong, J. and Kiniry, J. R.: EPIC and APEX: Model Use, Calibration, and Validation, Trans. ASABE, 55(4), 1447–1462, doi:10.13031/2013.42253, 2012.

White, C. M., Kemanian, A. R. and Kaye, J. P.: Implications of carbon saturation model structures for simulated nitrogen mineralization dynamics, Biogeosciences, 11(23), 6725–6738, doi:10.5194/bg-11-6725-2014, 2014.

Wutzler, T., Zaehle, S., Schrumpf, M., Ahrens, B. and Reichstein, M.: Adaptation of microbial resource allocation affects modelled long term soil organic matter and nutrient cycling, Soil Biol. Biochem., 115, 322–336, doi:10.1016/j.soilbio.2017.08.031, 2017.

Yang, X., Thornton, P. E., Ricciuto, D. M. and Post, W. M.: The role of phosphorus dynamics in tropical forests – a modeling study using CLM-CNP, Biogeosciences, 11(6), 1667–1681, doi:10.5194/bg-11-1667-2014, 2014.

Yu, L., Zanchi, G., Akselsson, C., Wallander, H. and Belyazid, S.: Modeling the forest phosphorus nutrition in a southwestern Swedish forest site, Ecol. Modell., 369, 88–100, doi:10.1016/j.ecolmodel.2017.12.018, 2018.

Yu, L., Ahrens, B., Wutzler, T., Schrumpf, M. and Zaehle, S.: Jena Soil Model (JSM

v1.0; Revision 1934): A microbial soil organic carbon model integrated with nitrogen and phosphorus processes, Geosci. Model Dev., 13(2), 783–803, doi:10.5194/gmd-13-783-2020, 2020.

Zhu, Q., Riley, W. J., Tang, J., Collier, N., Hoffman, F. M., Yang, X. and Bisht, G.: Representing Nitrogen, Phosphorus, and Carbon Interactions in the E3SM Land Model: Development and Global Benchmarking, J. Adv. Model. Earth Syst., 11(7), 2238–2258, doi:10.1029/2018MS001571, 2019.

---

## Author Comment (AC2) · 26 Jun 2020

Dear Dr. Kirkby, Thank you for your incisive review of the MS. We hope to address all your comments here. We will include your suggestions in the revised manuscript.

Dr. Clive Kirkby, Referee 2 (R2)

R2: "While the authors mention methodological and analytical discrepancies it is done in just a couple of lines. Do the authors think it is really important or not so important, and why. While it is pretty obvious why Pi is generally poorly correlated with soil C do the authors think soil C is poorly correlated with Po because of the unnamed methodological or analytical discrepancies or someother, perhaps unknown, reason."

Authors: We believe investigating the methodological and analytical discrepancies in

stoichiometry measurements to be an important pursuit. As stated in Kirkby et al. 2011, C:Po ratios can vary widely across studies (Table 7) employing different P measurement methods (extraction, digestion, or ignition). However, C:Po ratios also varied within studies applying the same methodology thus Kirkby et al. 2011 briefly explored some potential explanations for this weak correlation. We will address these mechanisms below and in more detail in the revised MS.

Kirkby et al. 2011 hints to varying Po forms and abundances as a possible driver of the variation in C to Po ratios: "... P is found in many organic compounds and the proportions vary in different soils (e.g. Turner et al., 2003a, 2003b). The major organic P compounds are the inositol phosphates (up to 50% of total OP), which contain neither N nor S but are generally considered to be associated with the soil heavy fraction component (Borie et al., 1989; Dalal, 1977)." This is a point we mention in line 180 and 366-67 of the review, on which we will expound more in the revised MS. Additionally, evidence of variable phytate abundance and mineralization is covered in section 2.1.3, but a more direct connection to the soil C to Po stoichiometry will also be made.

Another explanation mentioning diverging C:P ratios also appears in Kirkby et al. 2011: "differences in how microbes enzymatically attack OP vs N or S could also play a substantial role but we consider a detailed discussion on these issues beyond the scope of this paper". We comment on this in line 367-69 and more generally in section 2.1.3 and 3.1, but the relationship between enzymatic Po hydrolysis and soil stoichiometry will be stated more explicitly in section 2.2.

Mentioned later in our review are the differing mechanisms of stabilization of Pi and Po. Phosphate groups can control the sorption of Po, and some Po substances do not have to pass through the microbial biomass and can therefore be directly stabilized in the soil. An example of this would be phytate. Its recalcitrance and resulting abundance in some systems would yield a higher soil C:P ratio when compared to a similar system with a lower phytate abundance.

[Figure]

We will add more explicit connection between provided mechanisms and their effect on soil C:P stoichiometry: • Variable phytate abundance can affect SOM stoichiometry of soil. If more abundant in one soil, the C:P ratio could be higher than that of another with lower phytate abundance. • The prevalence of P-driven mineralization can affect the C:P ratio of the soil through preferential mineralization of P and not C.

Overall, it is difficult to ascribe a relative importance to each mechanism potentially controlling the decoupling of Po and C. However, we can describe the mechanisms that potentially explain this decoupling. When assessing if P cycling pathways or methodological difference explain this decoupling, we are inclined to say that both likely are at play (methods are described briefly in Kirkby et al. 2011 sec. 2.2), but that the review of P cycling mechanisms strongly suggests that no stability in this ratio should be expected. Rather, we should understand its controls.

R2: "While obtaining such a measurement is probably expecting too much it might be worthwhile seeing if forest soils and top soils in no till agriculture (which generally have high fungi:bacteria ratios) can be modelled differently to soils that are often cultivated (which often have lower fungi:bacteria ratios)."

Authors: It would be interesting to follow this line of thinking, and it would be natural to explore the scenarios proposed by the reviewer in a focused study. While we think that this falls outside the scope of the review, it does fall in the scope of topics that should be explored further and we will incorporate this suggestion it in the review.

R2: "The difference in fungal and bacterial C:P ratios may help to explain differences in P driven Po mineralization compared to C-driven Po mineralization in different soils"

Authors: Yes, this could be a very useful area of study and the relative abundance of fungi and bacteria (as R2 stated previously) is a worthwhile topic of investigation. Exploring fungi:bacteria ratios and the differences in their respective C:P ratios may reflect differences in P vs. C-driven mineralization and potentially, soil C:Po ratio discrepancies.

References

Borie, F., H. Zunino, and L. Martínez.: Macromolecule‐P associations and inositol phosphates in some Chilean volcanic soils of temperate regions, Communications in Soil Science and Plant Analysis 20(17-18), 1881-1894, 1989.

Dalal, R. C.: Soil Organic Phosphorus, Adv. Agron., 29(C), 83–117, doi:10.1016/S0065-2113(08)60216-3, 1977.

Turner, B. L., Cade-Menun, B. J. and Westermann, D. T.: Organic Phosphorus Composition and Potential Bioavailability in Semi-Arid Arable Soils of the Western United States, Soil Sci. Soc. Am. J., 67(4), 1168–1179, doi:10.2136/sssaj2003.1168, 2003a.

Turner, B. L., Chudek, J. A., Whitton, B. A. and Baxter, R.: Phosphorus composition of upland soils polluted by long-term atmospheric nitrogen deposition, Biogeochemistry, 65(2), 259–274, doi:10.1023/A:1026065719423, 2003b.

---

## Author Response (AR1)

Dear Dr. Sönke Zaehle,

Please accept this revised version of the manuscript bg-2020-130 titled "Reviews and Syntheses: Ironing Out Wrinkles in the Soil Phosphorus Cycling Paradigm" in addition to our response to the referees. Their comments were insightful, and we appreciate the input. We thoroughly considered all referee comments and clarified our text or added citations accordingly. In the marked-up manuscript, new text is highlighted while removed text is indicated by a strikethrough. In this document, we include comments from both referees, our responses, and the corresponding change in line number for both the clean and marked-up versions of the manuscript. Any line number referenced will be from the clean version, and the number following in parentheses corresponds to the marked-up document. e.g. Line 118(123).

The principal updates to the document are as follows: 1) Changed the language from "improving models" to "incorporating new concepts into models", which better represents the purpose of the paper. 2) Incorporated additional relevant work on modeling P biogeochemistry (most prominently P-driven $P_o$ mineralization). 3) Added other relevant citations recommended by referees, 4) Restructured paragraphs in sections 1.3, 1.5, and 2.2.1. 5) Clarified the difficulties in incorporating P-driven Po mineralization into models (section 1.5). Additionally, all instances of Pi, Po, and OC were changed to $P_i$, $P_o$, and $C_o$ respectively.

We look forward to your feedback. This letter, document, and revised manuscript represent the consensus of all co-authors.

Kind Regards,

Curt McConnell

Armen Kemanian

Jason Kaye

**Referee 1**

Dear reviewer, thank you for your incisive review of the MS. We hope to address all your comments here. The references that you provided are also pertinent and informative and will improve the manuscript.

**[Referee 1]:** "…it lacks in-depth insights of current "P models" to justify its statement that these processes should be improved in models. It is partly because some of the processes or features are simply ignored in current models, such as phytate, and partly because the proposed processes by authors are not particularly discussed by modeling community yet."

> **Authors**: The original intent in the initial stages of the review, was to compare individual models by reconstructing their respective P cycling components. That intent veered to the current manuscript with emphasis on missing components. As Referee 1 noted, there are many terrestrial biosphere models (TBMs) and landscape/watershed models. It was tedious and not necessarily informative to reconstruct them all due to the many overlapping frameworks and processes.
>
> With knowledge of individual models from this earlier research, we redirected our efforts to address concepts not generally or never included in these models (for example, phytate dynamics) or those that may not be representing a specific process completely (P-driven "biochemical" mineralization). We believe it is important to draw attention to those concepts ignored in models or not yet discussed by the modeling community, so that new hypotheses and experiments can be proposed, and models improved.
>
> In response to the comment regarding "improving P models", we think that the matter is more semantic than conceptual. We changed the language to "including processes not addressed in current models", which of course would be done to improve models. Our aims are to highlight areas of knowledge that need to be included in models in relation to P cycling, to acknowledge the need to converge on common or complementary frameworks well supported by theory, and to acknowledge that uncertainty still exists due to conceptual ambiguity or the inability to measure P pools and fluxes with ease.
>
> Revisions: Changed the language of "improving P models" to incorporating certain pools and processes or improving the modeling of P cycling generally.
>
> Line 436(459): "Regardless of the proposed mechanisms, in order to improve P modeling, accounting for competitive sorption reactions…"

**[Referee 1]:** "…'P cycling models', which in practice does not really count as a sub-category of models"

> **Authors**: The semantics here can be subjective. Any model refers to a system with arbitrary boundaries. Whatever comes from outside the boundaries are "forcings" or inputs. Any model "component", which can be called a submodel if inside a well bounded model, is in itself a model with its own forcings. Maybe we can make a stop here and highlight that is difficult to be a purist: setting boundaries does not make the boundaries correct, as there are feedbacks that could affect the forcing variables themselves. For example, a change in land use will affect the air temperature and air moisture through changes in the surface energy balance. For models that couple the atmosphere with the earth surface, climate variables are mostly forced in, regardless of the land use, and the feedback is ignored. The errors introduced by ignoring the feedback are in most cases minor. And of course there is no assurance that including a feedback will make the outcome better, it may just make the model more cumbersome veering into arrogance without any predictive gain. In any case, the model and sub-model language is context dependent. We will try to avoid any confusion.
>
> Revisions: Changed the language from "P models" to "P cycling" or "P components of biogeochemical models"
>
> Line 21(21): "Most P cycling components of biogeochemical models are structured after C and N…"
> Line 378(401): "Using fixed P stoichiometry to model P biogeochemistry may not capture P dynamics across ecosystems…"

**[Referee 1]:** "…perhaps the authors could try to narrow down the scope of models, for example, only to land surface models and TBMs"

>  Authors: Among the models referred to by Referee 1, it is important to realize that different names sometimes encapsulate the same or related P components. Referee 1 suggested including EPIC (Jones et al., 1984), GLEAMS (Leonard et al., 1987), and ANIMO (Schoumans and Groenendijk, 2000) in our review. GLEAMS was a standalone model. GLEAMS was absorbed into EPIC during the early stages of the EPIC model development. SWAT (as well as APEX) uses EPIC as the base crop and soil model for each simulation unit (SWAT is a semi-distributed watershed model and APEX is a pseudo-distributed model mostly applicable to smaller watersheds) (Gassman et al., 2007; Wang et al., 2012). By using SWAT, we were encompassing this conglomerate and trying to avoid long explanations and redundancy.

>  Revisions: Clarified the composition of SWAT and incorporated citations for CREAMS, GLEAMS, and EPIC

>  Line 523(547): "These equations were integrated into the Soil and Water Assessment Tool (SWAT) model (Arnold et al., 1998), which is built on the CREAMS, GLEAMS, and EPIC model structures (Knisel, 1980; Jones et al., 1984; Leonard et al., 1987)."

**Section 1:**
**[Referee 1]** in reference to Line 118 (original line #): "the modeling work has also supported that CP cycling is largely decoupled from CN cycling (Yu et al. 2020, GMD)"

>  **Authors**: We had missed the work of Yu et al. (2020) while finishing up the review. Indeed, Yu et al. 2020 is a good example of a model *representing* CP-CN decoupling. This reference will be included in our review. We agree with other comments in that work. The uncritical inclusion of a Century-like approach and the need to represent soil properties variation with depth as an emergent, and not an imposed property, has also been pointed out and partially addressed by using the carbon saturation concept (e.g. Kemanian and Stöckle, 2010), as so did Yu et al. (2020). We think that conceptually, Yu et al. (2020) and our review are convergent.

>  Revisions: Included Yu et al. 2020 as a work that represents the decoupling of CP-CN cycling.

>  Line 118(123): "A tighter coupling of C mineralization with soil organic matter (SOM) C:N ratios rather than C:P ratios (Heuck and Spohn, 2016), and a relatively poor correlation of Po with C or N (Yang and Post, 2011), indicate a greater independence of P mineralization from C than does N , a relationship captured by some simulation models (Yu et al., 2020)."

**[Referee 1]** in reference to Line 164 (original line **#):** "logically, I think it is better to place the last sentence at the beginning of this paragraph."

>  Revisions: Reordered paragraph as recommended

**[Referee 1]** in reference to Line 187 (original line #): "Schimel and Weintraub did not simulate phosphatase"

>  **Authors**: The intention was not to convey that Schimel and Weintraub (2003) used phosphatase activity in their theoretical model, rather to highlight that improving phosphatase activity assays would benefit model development employing approaches like that of Schimel and Weintraub, who account for enzymes in SOM decomposition. The text was changed as follows:

>  Revisions: Clarified the statement about the work of Schimel and Weintraub 2003.

>  Line 194(202): "Schimel and Weintraub (2003) took an enzyme-mediated approach to SOM decomposition/mineralization, but if applied to Po decomposition/mineralization, it would require more accurate phosphatase activity assays and an ability to differentiate between extracellular and intracellular soil phosphatase."

**[Referee 1]** in reference to Line 194 (original line #): "please check Yu et al. 2020, in which a dynamic enzyme allocation approach is used to mimic such a relationship"

> **Authors**: The point that we were making referred to measuring or explicitly modeling the relative release of phosphatases by both plants and microbes. The Yu et al. (2020) use of SEAM (Wutzler et al., 2017) and the ECA approach (Tang and Riley, 2013) addresses a different if related area, pertaining to the dynamic allocation of microbial phosphatases to litter or microbial residue and nutrient acquisition by plants and microbes.

> Revisions: Incorporated the work of Yu et al. 2020, Tang and Riley 2013, and Wutzler et al. 2017 to describe dynamic enzyme allocation, the ECA approach, and SEAM.

> Line 209(218): "…Yu et al. 2020 simulated Po mineralization by combining the soil enzyme allocation model (SEAM) (Wutzler et al. 2017) and the equilibrium chemistry approximation (ECA) (Tang and Riley 2013) to estimate the allocation of enzymes between Po sources and the subsequent microbial-plant uptake."

**[Referee 1]** in reference to Line 209 (original line #): "I would argue many terrestrial ecosystem models (ESMs) did include the P-driven mineralization. From as early as the CENTURY model (Parton et al. 1988) to the more recent ESMs such as JSBACH (Goll et al., 2012), ORCHIDEE (Goll et al., 2017), For SAFE (Yu et al., 2018), E3SM (Zhu et al., 2019), QUINCY (Thum et al., 2019). And the overall role of P-driven mineralization (biochemical mineralization) is crucial for plant growth in some of these models."

> **Authors**: We agree that many models include P-driven mineralization implicitly (CENTURY) (Achat et al., 2016) but those that do so explicitly (Goll et al., 2012, 2017; Yang et al., 2014) are parameterized on limited data (Reed et al., 2015). Goll et al. 2012 stated in relation to P-driven (biochemical) mineralization "parameter values for the adjustment intensity is set arbitrarily" and that in-field P-driven mineralization assays are not currently possible, thus limiting data. This is similarly stated by Thum et al. 2019 in reference to P-driven mineralization: "Further observations are required to constrain this flux better (Reed et al., 2011)." In other words, one can easily come up with a P flux from P-driven mineralization given P demand through a sort of inverse modeling. Our argument was that P-driven mineralization is not always explicitly included, and it is difficult to obtain P-driven mineralization as an emergent property. When P-driven mineralization is explicitly modeled, it is not well constrained. We could include more models into that categorization to make things clear and the review more thorough.

> Revisions: The works mentioned above were incorporated as examples of work that did simulate P-driven mineralization. We also added three methodical limitations and gaps in our understanding of P-driven mineralization:
> * Difficult to differentiate P-driven vs. C-driven mineralization
> * Phosphatase enzyme assays overestimate real phosphatase activity and cannot be conducted in the field
> * Difficult to relate phosphatase activity to mineralization

> Line 184-199(191-207): "Measuring and differentiating Po mineralization pathways is an important step for modeling P biogeochemistrythe P cycle. Some land surface and catchment models simulate P-driven Po mineralization (Wang et al., 2010; Goll et al., 2012, 2017; Davies et al., 2016; Yu et al., 2018, 2020; Thum et al., 2019; Zhu et al., 2019), but they are parameterized on limited observations (Reed et al., 2015; Thum et al., 2019), measured using unrepresentative or inaccurate methods. The first of these methodological limitations is the inaccuracy of P-driven mineralization measurements. It is difficult to quantify P-driven relative to C-driven mineralization with current assays (Oehl et al., 2004; Achat et al., 2016) particularly as phosphatase may play a role in both P and C-driven mineralization, complicating the use of phosphatase assays. Secondly, phosphatase enzyme assays are used as a proxy for P-driven mineralization (Goll et al., 2017), but the assays often overestimate real phosphatase activity and cannot be conducted in the field (Goll et al., 2012). Lastly, the mechanisms of phosphatase production and the relationship between phosphatase activity and mineralization is difficult to measure, and are therefore poorly understood and not explicitly simulated in models (Oehl et al., 2004; Achat et al., 2016). Schimel and Weintraub (2003) took an enzyme-mediated approach to SOM

decomposition/mineralization, but if applied to Po decomposition/mineralization, it would require more accurate phosphatase activity assays and an ability to differentiate between extracellular and intracellular soil phosphatase. Better representing the P cycle in models is less a question of calibration and more a need to improve our fundamental understanding of P and C-driven Po mineralization, which itself is hindered by the nature of P isotope chemistry, accessibility of appropriate methods, and currently held assumptions of the two pathways."

**Section 2:**
[Referee 1] in reference to Line 325 (original line #): "please check Lang et al. 2017, SBB."

> **Authors**: Lang et al. 2017 is a very fitting reference for the "potential shift to P-driven Po mineralization processes" statement (Line 358). The work by Lang et al. 2017 reflects the (Walker and Syers (1976), showing that as soils weather and overall P stocks degrade, Po becomes the more available P source as Pi is either lost from the system or occluded. Although not explicitly stated by Walker and Syers, McGill and Cole (1981) discussed the change in nutrient strategies with the depletion of P stocks: "Although Po has not started to decline, it has leveled off and the decline in total P is substantial. Under such conditions demand for internal cycling of Po to meet the needs for P increases." (McGill and Cole 1981)

> The objective of the paragraph in question referenced by Referee 1 was to describe how C:P ratios were variable and that they become more decoupled (wider range of possible C:P ratios) as soils weather. This may be due to the depletion in P stocks and changes in P nutrition strategies. It could also be due to variable C demand and the assumed prevalence of extracellular "P-driven 'biochemical' mineralization" to scavenge P. In a highly weathered soil where P availability is limiting, P could be scavenged without C mineralization, thus further decoupling C and P. Therefore, this decoupling would be dependent to some degree on the relative P to C demand and the extracellular mineralization of Po.

> Revisions: Included Lang et al. 2017 as their work on nutrient strategies fit well in this discussion.

> Line 333(355): "The decoupling of C:Pt and N:Pt is also seen as soil weathers (Yang and Post, 2011) where Po becomes the predominant contributor to P fertility (Yang and Post, 2011; Cleveland et al., 2013; Bünemann, 2015) and nutrient acquisition strategies shift from physiochemical Pi acquisition to closed Po cycling (Lang et al., 2017)."

[Referee 1] in reference to Line 366 (original line #): "The real challenge (for modeling) is to account for the huge stoichiometry differences between plant litter, microbe, and SOM, especially C:P ratio (Xu et al., 2013; Mooshammer et al., 2014). This requires an explicit microbial pool and a number of microbial adaptation processes to be included in models. Additionally, how to extrapolate these mechanisms from site-level to regional and global level is another complicated problem"

> **Authors**: We agree. Our intention was to convey this in the review while acknowledging that phytate dynamics are possibly related to the wide ratios in C and Po stoichiometry. The issues identified by the reviewer are on point but were also well understood. On the one hand, we highlight that the dynamics of phytate is barely in the modeler's radar. On the other hand, we acknowledge that phytate is just a component of the larger picture. We will rebalance the narrative.

> Revisions: Added paragraph stressing the importance of an explicit microbial pool and microbial adaptation processes.

> Line 385(408): "Including an explicit microbial pool that accounts for physiological and community dynamics (Allison, 2012; Wieder et al., 2014) when modeling the P cycle will better represent microbial adaptation to changing resource stoichiometry. However, further research into community specific drivers of microbial stoichiometry is needed, such as accounting for differences in the P acquisition strategies of bacteria and fungi (Waring et al., 2013). Representing this in models could be as simple as accounting for bacterial and fungal C:P or biomass ratios, akin to the approach of Waring et al. (2013) for C and N cycling, or a more complex trait-based approach (Allison, 2012; Wieder et al., 2014)."

**Section 3:**
**[Referee 1]** in reference to Line 398 (original line #): "there are some models implementing the OC saturation dynamics (or similarly clay-related C sorption capacity), such as MIMICS (Wieder et al. 2014), MEND (Wang et al. 2015), RESOM (Tang and Riley 2015), and COMISSION (Arhens et al. 2015). The recent JSM (Yu et al. 2020) has also include N and P in the OC sorption, which also complies with the saturation principle, although Po is not specifically separated as a competing sorbate of OC"

**Authors**: We are not certain we follow the reviewer, or if our point was clearly laid out and understood.

Wielder et al 2014 (MIMICS) include the effect of clay on SOM protection, not the concept of carbon saturation which is different (see Table 1 in that paper, row for $C_{scalar}$).

Wang et al 2013 and 2015 (MEND) does not address carbon saturation explicitly. There are parameters to represent C affinity to the mineral matrix, but no explicit mention of soil C saturation. Perhaps it is an emergent property of the model. However, the steady state mineral associated organic matter in equation 25 in Wang et al 2013 (variable M) does not seem to be saturating (when inputs go to infinity SOC should approach an asymptote to be "saturating"). That paper presents steady state equations, but not the limit when inputs go to infinity.

Tang and Riley (2015), figure 12b in the supplemental shows a linear increase in SOM with increasing mineral surface area (many models surrogate MSA with clay content despite the limitation of such simplification). There is however no indication of saturating behavior, although it might be an emergent property if the model is tested with increasing C inputs. It should also consider aggregation (see work by Six et al team, Kong et al., 2005), and Tang and Riley model does not address aggregation.

Ahrens et al. (2015) is explicitly saturating (Table 1, Eq 6 in their paper) and as acknowledged in that paper, it follows work by Hassink and Whitmore (1997). Ignoring fluxes between layers, the soil carbon steady state in Arhens et al is:

$$C_q = K_{ads} * q_{max} /(K_{ads} + k_{des}/C_{doc})$$

$C_q$ is soil organic carbon, $q_{max}$ is the maximum allowed soc, $C_{doc}$ is carbon in DOC, and $K_{ads}$ and $k_{des}$ are DOC sorption and desorption rates. As $C_{doc}$ goes to infinity $C_q = q_x$.

This is conceptually and mathematically similar to Hassink and Whitmore (1997) model, Kemanian et al (2006), Kemanian and Stockle (2010), and in a more formal formulation in Kemanian et al. (2011). A paper by White et al. (2014) in Biogeosciences followed up on the impacts on N dynamics. Arhens et al. (2015) is a good reference to include in this review, acknowledging that there are earlier precedents. And there is always the possibility that we are missing other relevant work.

We think that the general point that organic C saturation (not the response to clay) is not explicitly included in many models stands, as well as the need to consider the implications for P cycling, much as White et al. (2014) addressed the implications for N.

Revisions: No major changes were made, but Ahrens et al. 2015 was added, with justification mentioned above.

Line 419(442): "The dynamics of explicit $C_o$ saturation are only included in a few models (Kemanian and Stöckle, 2010; White et al., 2014; Ahrens et al., 2015), and its effect on Po mineralization is unexplored in models."

**[Referee 1]** in reference to Line 445 (original line #): "in principle, what is discussed in this paragraph makes sense, however, given the existing uncertainties in Pi sorption (as discussed in section 4), it is really a challenge ahead of our current focus to consider the role of Po and interactions with OC saturation. The role of OC in Pi sorption has been partially considered in QUINCY (Thum et al. 2019), which proves to have an important role in regulating the P availability and thus affect plant growth, but this is not specifically discussed in the model description paper."

**Authors**: Thank for the comment. It seems that Referee 1 agrees with the point. Stating the knowledge gap does neither ignore ongoing efforts to address it (even if they have not been published yet) nor the difficulties inherent to the problem.

**Section 4:**
**Referee 1 in reference to Line 501, 506 (original line #)**: "DI and STP not defined"

    **Authors**: Thank you. DI water = dionized water, STP = Soil test phosphorus

    Revisions: DI and STP were defined in the text.

    Line 521(545): "In lieu of a 24-hour incubation, known P concentrations were added to subsamples that were dried and rewetted three times with deionized (DI) water over a period of six months."

    Line 529(552): "However, solution P is assumed to be half of Mehlich-3 or other soil test phosphorus (STP) method."

**[Referee 1]** in reference to Line 517 (original line #): "The work by Helfenstein et al. 2020, BG, has shown that there is a certain level of correlation between the Hedley Pi pool and Isotopic Exchange Kinetics Pi pool. And our recent work actually shows, with the implementation of double-surface Langmuir on current Pi pool structure, there is a possibility of utilizing the Hedley data for model validation"

    **Authors**: An excellent point and a noteworthy inclusion in our review. However, the point we were making was that the numerous modifications of the Hedley method complicate the analysis of Hedley data. We did not intend to say that correlations can't be drawn, just that the lack of methodological consistency makes this difficult or contributes to a lingering uncertainty about the compatibility of chemically extracted and modeled P pools.

    Revisions: We included Helfenstein et al. 2020 as a citation in two locations where P turnover and transformations were being discussed. Below is the more relevant example.

    Line 541(565): "If we are to incorporate measurable pools into models, these pools need to be measured with a consistent protocol and efforts towards measuring P turnover and transformations using P radioisotope (Helfenstein et al., 2020) and oxygen isotope tracing (Joshi et al., 2016) must be continued."

**[Referee 1]** in reference to Line 525 (original line #): "Dari et al. 2015 not found in reference. And the effect of OC content on Pi sorption is already partially implemented in QUINCY (Thum et al. 2019, GMD)"

    **Authors**: Dari et al., 2015 reference is listed below. Its exclusion was unintentional.
    Dari, B., Nair, V. D., Colee, J., Harris, W. G. and Mylavarapu, R.: Estimation of phosphorus isotherm parameters: a simple and cost-effective procedure, Front. Environ. Sci., 3(October), 1–9, doi:10.3389/fenvs.2015.00070, 2015.

**[Referee 1]** in reference to Figure 1: "Isn't the weathering of P coming from apatite P? are the three Pi and Po pools forming a continuum of stability? Do they transfer with each other? If they do, please add internal fluxes between pools. If they don't, do they all directly transfer phosphate to solution Pi?"

    **Authors**: Yes, apatite P is the most prevalent source, the use of "P minerals" was purposefully generic, and referring to it as apatite seems to be most common in the literature. There is likely exchange between the various pools through stabilization and destabilization as mineral-$PO_4$ bonds are formed and broken. However, there should be direct exchange between all soil pools and the soil solution. This is particularly pertinent to organic phosphates. There may be more "stable" Po forms, such as phytate, but P can be hydrolyzed using specific phosphatases such as phytase. P from any Po pool can be directly hydrolyzed into the soil solution, albeit at different rates depending on the stability of the Po form under consideration.

Actual "pools" are less well-defined than those operationally defined by chemical extractions. But for clarity, we will add arrows indicating internal fluxes between Pi and Po pools. All pools can directly transfer to the solution pool; P does not have to cascade down the gradient of pool stability until it reaches the "solution" pool. It is likely that all pools exchange P with the solution and with one another. Representing this will increase the number of arrows and complexity of diagram, but will make it more representative.

Revisions:
1. "Apatite" weathering replaced "Weathering of P minerals"
2. Arrows were added to indicate exchange between P pools

**[Referee 1]** in reference to Figure 3: "Are there any references for the C:P ratios values in the figure?"

**Authors**: Value (description) Reference. Microbial biomass (MB). Threshold Elemental Ratio (TER)

25 (MB C:P ratio, low of range) Capek et al. 2016;
186 (Soil C:P ratio, global average) Cleveland et al. 2007;
350 (MB C:P ratio, high of range) Capek et al. 2016;
1400 (Threshold Element Ratio), Heuck et al. 2016;
4000 (Decomposition stop point, Zechmeister-Boltenstern et al. 2015)

Revisions:
The references were added to Figure 3.

**[Referee 1]** in reference to Figure 6: "the resolution of the figure is too low to read. What are the differences between the two dash lines in each sub-panel?"

**Authors**: We improved the resolution. The dashed lines are regressions for all the Land Use/Vegetation categories in the study. The dotted lines are regressions for Land Use/Vegetation whose slope was significantly different than other systems (e.g. the "Wetland Organic" slope is negative in the top-left regression). We could describe which system regressions differ from the rest in the figure legend if that improves clarity. The important point is that the $r^2$ values are lower for C:P in soils and microbes than the C:N.

Revisions:
The resolution problem of Figure 6 was fixed. The legend is as follows:
"Figure 6. Regressions of C:N (A), C:P (B), and N:P (C) ratios in soils and the microbial biomass using a log10 transformation. The dashed lines (standard font $r^2$) are regressions of soil or microbial biomass stoichiometry for all land use/vegetation types that do not differ significantly. The dotted lines are slopes of stoichiometries that differ significantly from the other land use/vegetation types. For C:N (A), these land use/vegetation types were wetland organic, boreal forest, and humic horizons. For C:P (B) and N:P (C) the forest and pasture soils differed significantly from the other land/use vegetation types. The solid lines represent the Redfield (1958) ratio. Modified from Hartman and Richardson 2013."

**Referee 1 in reference to Table 2**: "it needs an appropriate caption. There are quite some different equations for C sorption capacity and PSP, why do you choose those specific ones. I am not particularly sure about the purpose of these equations here, particularly the ones calculating PSP."

Authors: PSP equations are listed because they were described earlier in the text. Showing the equations makes the model structure unequivocal.

**Referee 2**

Dear Dr. Kirkby, Thank you for your incisive review of the MS. We hope to address all your comments here. We will include your suggestions in the revised manuscript.

**[Referee 2]** in reference to C:P variability: "While the authors mention methodological and analytical discrepancies it is done in just a couple of lines. Do the authors think it is really important or not so important, and why. While it is pretty obvious why Pi is generally poorly correlated with soil C do the authors think soil C is poorly correlated with Po because of the unnamed methodological or analytical discrepancies or some other, perhaps unknown, reason."

> **Authors**: We believe investigating the methodological and analytical discrepancies in stoichiometry measurements to be an important pursuit. As stated in Kirkby et al. 2011, C:Po ratios can vary widely across studies (Table 7) employing different P measurement methods (extraction, digestion, or ignition). However, C:Po ratios also varied within studies applying the same methodology thus Kirkby et al. 2011 briefly explored some potential explanations for this weak correlation. We address these mechanisms below.
>
> Revisions: We moved the first paragraph of 2.2.1 to 2.2 in order to better introduce the methodological vs. mechanical explanation for variation in soil and microbial stoichiometry. We also added the following to the of the paragraph in 2.2 to outline the drivers of variable soil and microbial C:P stoichiometry:
>
> Line 324(341): "Variability in soil and microbial stoichiometry derive from methodological or analytical discrepancies (Kirkby et al., 2011), edaphic and ecosystem properties, and microorganism-specific characteristics (Mooshammer et al., 2014; Čapek et al., 2016), all of which must be considered when modeling plant litter and SOM decomposition."
>
> (1) Kirkby et al. 2011 hints to varying Po forms and abundances as a possible driver of the variation in C to Po ratios: "… P is found in many organic compounds and the proportions vary in different soils (e.g. Turner et al., 2003a, 2003b). The major organic P compounds are the inositol phosphates (up to 50% of total OP), which contain neither N nor S but are generally considered to be associated with the soil heavy fraction component (Borie et al., 1989; Dalal, 1977)."
>
> Revisions: We mention Po forms and their varying recalcitrance as a potential explanation to C:P decoupling (Line 330), briefly expounding on this at Line 358. The evidence of variable phytate abundance and mineralization is already outlined in section 2.1.3.
>
> Line 336(358): "Another possibility for this stoichiometric decoupling is the varying abundance of specific Po forms between systems (Kirkby et al., 2011). Because the C:P ratio of phytate is 1, shifting or variable phytate abundance would reflect in the soil C:P stoichiometry."
>
> Mentioned later in our review are the differing mechanisms of stabilization of Pi and Po. Phosphate groups can control the sorption of Po, and some Po substances do not have to pass through the microbial biomass and can therefore be directly stabilized in the soil. An example of this would be phytate. Its recalcitrance and resulting abundance in some systems would yield a higher soil C:P ratio when compared to a similar system with a lower phytate abundance.
>
> (2) Another explanation mentioning diverging C:P ratios also appears in Kirkby et al. 2011: "differences in how microbes enzymatically attack OP vs N or S could also play a substantial role but we consider a detailed discussion on these issues beyond the scope of this paper".
>
> Revisions: We mentioned this in the review at line 360: "A depletion of mineral Pi, an absence of a strong C limitation, changes in abundance of certain Po forms, or a potential shift to P-driven Po mineralization processes, may explain this decoupling." This was again mentioned again in a revised statement at line 405.

Line 383(405): "and second, P-driven Po mineralization can drive the decoupling of P mineralization from SOM decomposition (Goll et al., 2012)"

Overall, it is difficult to ascribe a relative importance to each mechanism potentially controlling the decoupling of Po and C. However, we describe the mechanisms that potentially explain this decoupling. When assessing if P cycling pathways or methodological difference explain this decoupling, we are inclined to say that both likely are at play (methods are described briefly in Kirkby et al. 2011 sec. 2.2), but that the review of P cycling mechanisms strongly suggests that no stability in this ratio should be expected. Rather, we should understand its controls.

[Referee 2]: "While obtaining such a measurement is probably expecting too much it might be worthwhile seeing if forest soils and top soils in no till agriculture (which generally have high fungi:bacteria ratios) can be modelled differently to soils that are often cultivated (which often have lower fungi:bacteria ratios)."

Authors: It would be interesting to follow this line of thinking, and it would be natural to explore the scenarios proposed by the reviewer in a focused study. While we think that this falls outside the scope of the review, it does fall in the scope of topics that should be explored further and we will incorporate this suggestion it in the review.

Revisions: Research on fungi:bacterial C:P or biomass ratios was added to the paper to address the reviewer comment. See following comment.

[Referee 2]: "The difference in fungal and bacterial C:P ratios may help to explain differences in P driven Po mineralization compared to C-driven Po mineralization in different soils"

Authors: Yes, this could be a very useful area of study and the relative abundance of fungi and bacteria (as Referee 2 stated previously) is a worthwhile topic of investigation. Exploring fungi:bacteria ratios and the differences in their respective C:P ratios may reflect differences in P vs. C-driven mineralization and potentially, soil C:Po ratio discrepancies.

Revisions: Added paragraph briefly describing C:P decoupling and microbial adaptation and how accounting for fungal vs. bacterial stoichiometry may help modeling efforts.

Line 387(410): "further research into community specific drivers of microbial stoichiometry is needed, such as accounting for differences in the P acquisition strategies of bacteria and fungi (Waring et al., 2013). Representing this in models could be as simple as accounting for bacterial and fungal C:P or biomass ratios, akin to the approach of Waring et al. (2013) for C and N cycling, or a more complex trait-based approach (Allison, 2012; Wieder et al., 2014)."

**Citations added and justification**
General Papers:
- (Helfenstein et al., 2020): Included because the work uses P radioisotopes to study turnover of Hedley Fractions, a concept mentioned in the text.
- (Lang et al., 2017): Added in response to author comment as the work explores the nutrient strategies of plants/microbes as P stocks change.
- (Mooshammer et al., 2014): Added because it adds to the discussion of the non-strictly homeostatic nature of microbes. Review covering microbial/soil stoichiometry.
Models added:
- (Ahrens et al., 2015): Added as an example of a model that represents C saturation.
- (Allison, 2012): Added as an example of a trait-based model of litter decomposition.
- (Goll et al., 2017): Added as an example of a model that represents P-driven mineralization, but one that uses phosphatase activity as a proxy for P-driven mineralization.
- (Davies et al. 2016): Added as an example of a model that represents biochemical mineralization

- (Knisel, 1980; Jones et al., 1984; Leonard et al., 1987): All components of SWAT (GLEAMS, EPIC, and CREAMS).
- (Tang and Riley, 2013): Cited when describing the dynamic enzyme allocation approach of Yu et al. 2020.
- (Thum et al., 2019): Added as the paper supports the claim that biogeochemical models representing biochemical mineralization of P are parameterized on limited observations
- (Waring et al., 2013): Accounts for fungal:bacterial ratios in model (a recommended approach by Referee 2)
- (Wieder et al., 2014): Included because the work calls for explicitly modeling microbial processes.
- (Wutzler et al., 2017): Cited as a component of the dynamic enzyme allocation approach of Yu et al. 2020.
- (Yu et al., 2018): Included as model that represents P-driven Po mineralization.
- (Yu et al., 2020): Included as a model that represents CP decoupling from CN.
- (Zhu et al., 2019): Included as model that represents P-driven Po mineralization.

[revised manuscript text omitted]

---

## Referee Report (RR1)

Line 35: nutrient models -> biogeochemical models.

Line 43: overview of SOIL P cycle. Also, many appearances of "P cycle" in the paper should be changed to "soil P cycle"

Line 61: please check the usage of "availability" in the paper. Since the authors haven't mentioned bioavailability, so each appearance of "availability" is slightly different due to the context? For example "labile Pi availability" (or "labile Pi content"???), "mineral P availability", "Pi availability", and "phytate availability for plant acquisition". I would recommend change the wording in some circumstances to avoid confusion.

Line 377: "P cycling models"->"models"

Line 437: the statement is too strong, and "in order to improve P modeling" sounds a bit odd.

Line 566: remove the strikethrough line in "P̶ simulation models"

---

## Author Response (AR2)

Dear Dr. Sönke Zaehle,

Please accept this minor revision of the manuscript bg-2020-130 titled "Reviews and Syntheses: Ironing Out Wrinkles in the Soil Phosphorus Cycling Paradigm". In the marked-up manuscript, new text is highlighted while removed text is indicated by a strikethrough and highlighted in red. In this document, we include your comments, our responses, and the corresponding change in line number for both the clean and marked-up versions of the manuscript. Any line number referenced will be from the clean version, and the number following in parentheses corresponds to the marked-up document. e.g. Line 118(123).

We carefully addressed your line-specific comments in addition to your general grammar recommendations. Grammatical corrections primarily consisted of removing extraneous commas. Additionally, we reduced sentence length in some instances and removed or added a small number of words to enhance clarity. Only one sentence was added in order to provide more context for a paragraph in section 3.3, paragraph 4. Lastly, Figure 1 was corrected as a pair of brackets was accidentally omitted in the first revision.

We look forward to your feedback. This letter, document, and revised manuscript represent the consensus of all co-authors.

Kind Regards,

Curt McConnell

Armen Kemanian

Jason Kaye

**Associate Editor [AE]:**

Dear authors,

many thanks for your revised manuscript. After review of one of the original reviewers and my own assessment, I am satisfied with the way you have responded to the criticims raised. Both reviewer #1 and I have some minor technical issues that you should resolve before this manuscript can be published in Biogeosciences. I also found a smaller number of misplaced commas or points. Therefore, I would like to ask you to carefully check spelling and punctuation again, including a consideration of splitting longer sentences for better readibility.

Best wishes,
Sönke Zaehle

Editorial comments:
L29: clarify what "regarding P" means.
L134: no new paragraph
L295: Give section number
L323: remove quotation marks, or explain.

**Corrections:**

**[AE] L29:** clarify what "regarding P" means.
> Line 29(30): "cycling" was added after P as we were referring to results of models of the P cycle.

**[AE] L134:** no new paragraph
> Line 132(134): Paragraphs were combined

**[AE] L295:** Give section number
> Line 292(295): The section number was provided in this instance, and any other instance where a section was referenced.

**[AE] L323:** remove quotation marks, or explain.
> Line 320(323): Removed quotations and added source (already previously cited) that uses the "non strictly homeostatic behavior" wording.

**Other minor edits:**

Line 460(463): Added to provide some more context for the subsection: "The accumulation of Po has implications for how DPS measurements of $P_i$ are in interpreted; $P_o$ can competitively sorb with $P_i$ and reduce the measured, but not actual, DPS"

Line 455(458): Changed "C1" to "Figure 7C" to make this an appropriate figure reference.

Figure 1: The brackets around the inorganic phosphate pools (gray boxes) were accidentally omitted in the first revision due to a formatting error. This was error was corrected and the brackets restored.

**Changes in grammar and sentence structure:**

Split sentences for clarity:
Lines: 25-27(26-7); 85-87(86-8); 195-197(197-99); 432-434(435-37)

Minor word change for grammar or clarity
Example Lines: 129(131); 318(321)

[revised manuscript text omitted]